# Cryo-EM structures of pannexin 1 and 3 reveal differences among pannexin isoforms

Nazia Hussain [1], Ashish Apotikar [1], Shabareesh Pidathala [1,3], Sourajit Mukherjee [1,4], Ananth Prasad Burada [1], Sujit Kumar Sikdar [1], Kutti R. Vinothkumar [2] & Aravind Penmatsa [1] ✉

Pannexins are single-membrane large-pore channels that release ions and ATP upon activation. Three isoforms of pannexins 1, 2, and 3, perform diverse cellular roles and differ in their pore lining residues. In this study, we report the cryo-EM structure of pannexin 3 at 3.9 Å and analyze its structural differences with pannexin isoforms 1 and 2. The pannexin 3 vestibule has two distinct chambers and a wider pore radius in comparison to pannexins 1 and 2. We further report two cryo-EM structures of pannexin 1, with pore substitutions W74R/R75D that mimic the pore lining residues of pannexin 2 and a germline mutant of pannexin 1, R217H at resolutions of 3.2 Å and 3.9 Å, respectively. Substitution of cationic residues in the vestibule of pannexin 1 results in reduced ATP interaction propensities to the channel. The germline mutant R217H in transmembrane helix 3 (TM3), leads to a partially constricted pore, reduced ATP interaction and weakened voltage sensitivity. The study compares the three pannexin isoform structures, the effects of substitutions of pore and vestibule-lining residues and allosteric effects of a pathological substitution on channel structure and function thereby enhancing our understanding of this vital group of ATP-release channels.

Pannexins are large-pore vertebrate ion channels identified through sequence similarity with invertebrate gap-junction channels, innexins[1]. Pannexin protomers retain structural and topological similarity to innexins, connexins, and volume-regulated anion channels but remain as single-membrane channels and participate in ion-conductance and ATP release[2]. Three isoforms of pannexins (PANX1, 2 & 3) are identified in mammalian cell types, and each is observed in diverse tissue and physiological niches[3].

Pannexin 1 (PANX1) is the more extensively studied member among pannexins, observed ubiquitously, and is involved in purinergic signaling through ATP release[4]. It is documented to play a role in diverse processes, including inflammation, cell migration, apoptosis, cytokine secretion, and viral replication[5–8]. The C-terminus of PANX1 is susceptible to caspase 3/7 cleavage that results in channel-opening and the released ATP acts as a "find-me" signal for macrophages to target

apoptotic cells[9]. Besides ATP release, PANX1 is also known to display ionic conductance in response to extracellular potassium, positive potentials, and in some instances display mechanosensitive properties[10,11]. PANX1 knockout results in diverse phenotypes like hearing loss, reduced seizure activity, and reduced ATP/cytokine release[12–14]. A recent identification of a germline mutant (R217H) of *Panx1* that leads to defective channel activity resulting in intellectual disability, ovarian failure, hearing loss, and skeletal defects[15] signifies the importance of PANX1 as an essential player in human physiology.

A lot less is known about the two other isoforms of PANX1, namely PANX2 and PANX3, which have tissue-specific expression patterns. PANX2 is primarily expressed in neurons and glial cells and is involved in cellular differentiation[16]. Recent structural studies of PANX2 have further highlighted the unique properties of this isoform[17,18]. PANX3 is expressed in osteoblasts, chondrocytes, and skin and plays a major

[1]Molecular Biophysics Unit, Indian Institute of Science, Bangalore 560012, India. [2]National Centre for Biological Sciences, Tata Institute of Fundamental Research, Bangalore 560065, India. [3]Present address: St. Jude Children's Research Hospital, Memphis, TN, USA. [4]Present address: Department of Chemistry, The University of Chicago, Chicago, USA. ✉e-mail: penmatsa@iisc.ac.in

role in calcium homeostasis suggesting an independent functional niche[19]. Despite the high sequence conservation with its isoform, PANX3 displays distinct structural features, localization, and functional characteristics compared to PANX1[20]. PANX3 has been observed to regulate osteoblast differentiation through its ER calcium channel function[21]. Along with its predominant role in osteoblast differentiation and wound healing in mice[22], PANX3 also functions as an ATP release channel similar to PANX1 and PANX2, and the released ATP acts as an activator for P2R-PI3K-AKT signaling[21]. PANX3 shares 42% sequence identity with PANX1 and is the shortest among the three isoforms with a length of 392 residues[3]. Despite the differences, PANX3 is suggested to functionally compensate for the absence of PANX1 in the vomeronasal organ of PANX1 knock out mice[23]. Multiple attempts to observe PANX3 channel activity in different cellular contexts have been unsuccessful[24,25]. PANX2 is the most divergent among the three, with a sequence identity of 27% with PANX1, and is characterized by a substantially longer C-terminus[3]. The PANX isoforms, PANX1, PANX2, and PANX3 harbor substitutions in the residues that control channel gating and can have architectural differences that form the basis for altered properties among the three isoforms.

Recent structural studies on different orthologues of pannexin 1 and 2 (PANX1, PANX2) have shown that the channel is organized as a heptamer[17,26–28]. The topology is similar to large-pore ion channels such as calcium homeostasis modulators (CALHMs), leucine rich repeat containing VRAC subunit 8 (LRRC8), connexins and innexins with four transmembrane helices (TMs). The channels also display a prominent extracellular domain (ECD), intracellular domain (ICD) and a disordered C-terminus that is not visible in most PANX1 structures[29]. The access to the channel vestibule for the C-terminally truncated PANX1 and 2 isoforms through the cytosol is unhindered except for a constriction point at the ECD that would enforce ion selectivity/ATP gating depending on the pore radius. This pore, lined by tryptophan (W74) in PANX1 and arginine (R89) in PANX2 in loop1, is a crucial site for channel gating and inhibitor (carbenoxolone, CBX) interactions that block PANX1[25,26]. Incidentally the PANX3 has a branched hydrophobic amino acid, isoleucine (I74), at this position. These isoform-specific changes at the pore can influence channel properties and we set out to explore the structural and functional effects of these pore lining residues in pannexin isoforms 1 and 3. Here, we present the electron cryo-microscopy (cryo-EM) structures of human pannexin 3 (PANX3) and mutants of human pannexin 1 (PANX1) to observe effects on channel organization, pore radii and effects on ATP interactions. The PANX3 structure displays a constriction point below the pore that facilitates the formation of a second vestibular cavity in PANX3. We observe that the germline mutant R217H (PANX1$_{R217H}$) in TM3, in comparison with PANX1$_{WT}$, results in the allosteric modulation of the PANX1 pore to constrict it further and significantly alter its conductance properties. Cationic residue substitutions at distinct locations within the channel vestibule along the ATP permeation pathway ranging from the cytosolic face of PANX1 to the pore lining residues yielded altered affinities for ATP analog. The study attempts to comprehensively analyze the structure of PANX3 in comparison to other isoforms and demonstrate alterations in channel gating in PANX1 as a consequence of critical mutations.

## Results

### Pannexin 3 has distinct structural features in comparison to other isoforms

The PANX3 isoform shares 42% sequence identity with PANX1 (Supplementary Fig. 1) and 27% identity with PANX2. To understand the architectural differences among PANX isoforms, we purified the full-length human PANX1 and 3 isoforms and elucidated the cryo-EM structures to an overall resolution of 3.75 Å and 3.9 Å, respectively (Supplementary Figs. 2 and 3) The PANX3 structure was determined in the presence of ATP (1 mM) and high K$^+$ (100 mM; Fig. 1a, b, Supplementary Fig. 3, and Table 1). Similar to PANX1, high extracellular

potassium is also speculated to open PANX3 channels[30]. We used 100 mM K$^+$ during final purification of both PANX1 and PANX3 in an attempt to capture an open conformation of the channels. While we successfully determined the structure of PANX1$_{WT}$, for structural comparison, we utilized previously published PANX1 (PDB ID-6WBF) structure coordinates due to its higher resolution[26]. The density at the pore and transmembrane helices in PANX3 was sufficient to model most of the side chains in these regions (Supplementary Fig. 4). The intracellular helices (160-185) and the C-terminus (373-392) lack clear densities, likely due to inherent flexibility in this region. Density for bound ATP molecules that would allow unambiguous assignment, at this resolution, was not observed. The PANX3 retains the heptameric oligomer assembly that can be partitioned into an extracellular domain (ECD), transmembrane domain (TMD), and intracellular domain (ICD), similar to PANX1$_{WT}$ and PANX2 (Fig. 1a, b). Unlike other large-pore ion-channels like CALHMs[31], the PANX isoforms do not display heterogenous oligomeric associations[32]. The PANX3 channel is 8 Å wider than the PANX1 at the cytosolic face with similar transmembrane length in the ordered regions of the channel (Fig. 1c). The channel width of PANX3 at the extra and intracellular faces is similar to PANX2 (Fig. 1c).

The topology of PANX3 protomers is similar to PANX1$_{WT}$ with differences in the TM1 and the extracellular loops. The topology comprises four transmembrane helices and two extracellular loops (EL). Two disulfide bonds (SS1 and SS2) stabilize the extracellular domain, C66-C261(SS1) and C84-C242 (SS2), that form between EL1 and EL2 (Supplementary Fig. 5a, b). The superposition of two protomers (PANX1 vs. PANX3) yields a root mean square deviation (rmsd) of 3.2 Å for the 302 Cα atoms aligned. There is also a significant alteration of surface charge within PANX3 compared to PANX1$_{WT}$ and PANX2 near the surface lining the solvent-accessible channel vestibule (Fig. 1d).

N-linked glycosylation at the N255 position in the EL2 of PANX1 was implicated in preventing the formation of gap junctions[26,33]. A substitution at this site (N255A) led to the formation of a mixture of gap junctions and hemichannels[26]. In the PANX3 structure, we observed density for N-acetylglucosamine (NAG) at the predicted glycosylation site (N71) in the first extracellular loop that is much closer to the pore, in comparison to PANX1 (Supplementary Fig. 5c, d). The N-glycosylation in PANX2 is also localized to the EL1 region at N86 position that coincides with PANX3 *N*-glycosylation site (Supplementary Fig. 1) although the structure of PANX2 in a recent report does not reveal the site due to a likely disorder in the vicinity[17].

In PANX3, we do not observe density for residues 1-24 in N-terminus. It is therefore unclear if the N-terminus lines the pore and plays a role in maintaining the rigidity of the transmembrane domain of the heptamer similar to PANX1. Towards the C-terminus, we did not observe the density for the last twenty residues (373–392). The C-terminus is comparatively shorter than the corresponding stretch of PANX1 and lacks a caspase cleavage site to facilitate ATP release, as observed in PANX1. Moreover, ATP release has been observed in PANX3 in presence of high extracellular potassium even without the deletion of C-terminus of PANX3[30]. To test the functional effects of the C-terminus in affecting channel properties, we designed a deletion construct (PANX3$_{C-del}$) of PANX3 lacking the last 22 residues (372–392). We performed electrophysiology experiments and found no discernible change in current associated with the deletion of the C-terminus when compared with PANX3$_{WT}$, although the surface expression of the deletion mutant was higher than PANX3$_{WT}$ (Supplementary Fig. 6a, b). This observation led us to suggest that the short C-terminus of PANX3 may not play a significant role in channel opening and PANX3 may have alternate mechanisms for channel-gating.

### PANX3 has a wider pore among pannexins, and pore mutants alter channel properties

The residue lining the constriction point in PANX1, W74, is replaced by isoleucine or valine in different orthologues of PANX3 (Fig. 2a, b and

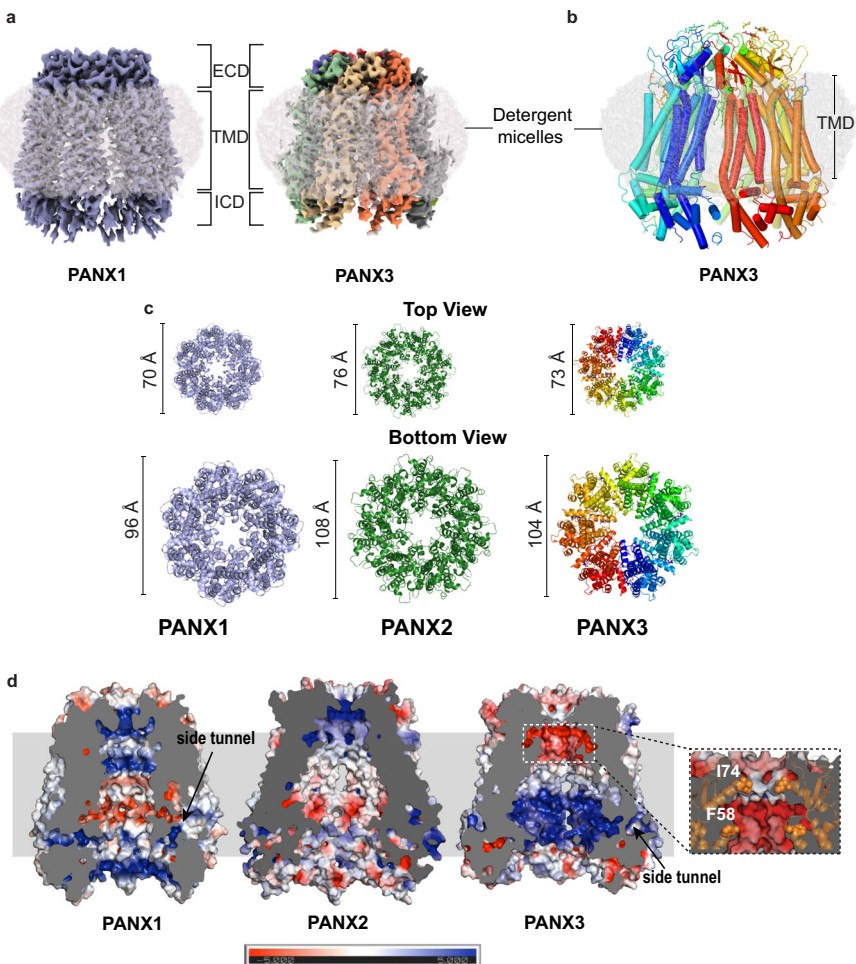

**Fig. 1 | Structural comparison of PANX3 with other isoforms. a** The cryo-EM map for PANX1 (blue) and PANX3 (protomers colored individually) viewed parallel to the membrane plane and surrounded by the detergent micelle (gray). **b** The modeled structure of PANX3 displaying heptameric organization and embedded in a detergent micelle that marks the transmembrane boundary. **c** Top (extracellular) and bottom (cytosolic) views of PANX1, 2 and PANX3 exhibiting differences in the dimensions in PANX isoforms at the extracellular and intracellular faces of the channel. **d** Sagittal section of surface electrostatics of PANX1 (PDB ID: 6WBF), PANX2 (PDB ID: 7XLB), and PANX3 colored according to potential from −5 (red) to +5 (blue) ($k_BTe_c^{-1}$) viewed parallel to the membrane plane. The inset shows the position of the residues I74 and F58 (orange spheres) forming first and second constrictions, respectively in PANX3.

Supplementary Fig. 6c). The human PANX3's pore is lined by two residues, I74 and R75, with a width of 13.2 Å at the narrowest point created by I74 (Fig. 2b). The primary constriction created by the W74 residue in PANX1$_{WT}$ results in a pore diameter of 12 Å. The presence of I74 in PANX3 instead of W74 widens the pore to a diameter of 13.2 Å. The cation-π interaction between W74 and R75 in PANX1$_{WT}$ is eliminated in PANX3 due to the replacement of W74 with I74. R75 of one PANX3 protomer interacts with D81 of the other protomer through a salt bridge (Fig. 2d, f, h). Residues in the same position in the pore in PANX2 are substituted with R89 and D90 that are drastically different from residues observed in PANX1 and PANX3. R89 in PANX2 is suggested to form highly cationic pore at the channel entrance (Fig. 2e). However, unlike PANX1 and PANX3, the main pore residues, R89 and D90 in PANX2, do not engage in interprotomeric interactions (Fig. 2e, g)[17].

Moreover, S70 and Q76 between two protomers of PANX3 form a hydrogen bond resulting in interprotomeric interactions leading to a stable heptamer. Apart from these interactions, there seem to be minimal intersubunit interactions across the ECD and TMD regions between the protomers (Supplementary Fig. 6e). The gap between the TMs 2 and 4 of adjacent protomers just beneath the ECD is occupied by lipid-like density into which we have modeled a phospholipid corresponding to the upper leaflet of the bilayer, 1-palmitoyl-2-oleoylphosphatidylethanolamine (POPE), for each protomer

(Supplementary Fig. 5e). The phospholipid could be enhancing the interactions between the protomers of PANX3 akin to the phospholipid interactions observed in the CALHM1 channels[34]. However, besides this lipid density we did not observe any other prominent density for lipid at the inner leaflet of the bilayer likely due to the lower resolution of this structure compared to other channels among large-pore ion channels. As a consequence, we could observe a gap between subunits that is large enough to allow the passage of ions. The presence of a side-tunnel suggested by Ruan et al. in PANX1$_{WT}$ is prominent in PANX3 with a 6.9 Å separation between TM2 and CTH1 compared to 6.3 Å in PANX1$_{WT}$, suggesting that the lateral portal hypothesis may also hold true for PANX3 (Supplementary Fig. 6f). Although we did not observe a lipid density around this region that could occlude this site, a higher resolution structure of PANX3 might reveal greater details about the solvent accessibility around this portal. This region is occluded with a distance of 2.6 Å between TM2 and CTH1 in the case of PANX2 and is unlikely to support the entrance of ions from this portal in PANX2 in the conformation, reported from the model (Supplementary Fig. 6f).

The structural comparison of PANX3 with PANX1$_{WT}$ reveals an additional constriction in PANX3 below the primary channel pore. The residues at the end of TM1, 58-60 comprising residues F58, S59, and S60 form a prominent second constriction at the neck region in PANX3 compared to PANX1$_{WT}$ (Fig. 2a, c). The linker between TM1 and TM2

**Table 1 | Cryo-EM data collection, refinement, and validation statistics**

| | PANX1$_{WT}$ (EMD-34268) | PANX1$_{WR/RD}$ (EMD-34267) (PDB 8GTT) | PANX1$_{R217H}$ (EMDB-34266) (PDB 8GTS) | PANX3 (EMDB-34265) (PDB 8GTR) |
|---|---|---|---|---|
| Data collection and processing | | | | |
| Magnification | 75,000 | 105,000 | 130,000 | 130,000 |
| Mode | TEM | EFTEM | EFTEM | EFTEM |
| Voltage (kV) | 300 | 300 | 300 | 300 |
| Detector | FalconIII | Gatan K3 | Gatan K2 | Gatan K2 |
| Slit width (eV) | – | 20 | 20 | 20 |
| Electron exposure (e⁻/Å²) | 29.5 | 39.84 | 41.12 | 48.8 |
| Defocus range (μm) | −1.8 to −3.3 | −1 to −2.4 | −1.8 to −3.3 | −1.8 to −3.3 |
| Pixel size (Å) | 1.07 | 0.84 | 1.07 | 1.07 |
| Micrographs (No.) | 1201 | 15500 | 1776 | 1112 |
| Symmetry imposed | C7 | C7 | C7 | C7 |
| Initial particle images (no.) | 399,706 | 3,869,579 | 781,859 | 411,263 |
| Final particle images (no.) | 41483 | 80029 | 40873 | 31517 |
| Map resolution (Å) | 3.75 | 3.20 | 3.87 | 3.91 |
| FSC threshold | 0.143 | 0.143 | 0.143 | 0.143 |
| Refinement | | | | |
| Initial model used (PDB code) | | 6WBF | 6WBF | Alphafold2 model |
| Model resolution (Å) @ FSC 0.5 | | 3.5 | 4.2 | 4.2 |
| Map sharpening *B* factor (Å²) | −179.6 | −106.6 | −156.0 | −143.6 |
| Model composition | | | | |
| Non-hydrogen atoms | | 16247 | 15470 | 16975 |
| Protein residues | | 2121 | 2016 | 2156 |
| Ligands | | – | – | 7 (NAG) 7 (PTY) |
| B-factor (Å²) | | | | |
| Total | | 67.97 | 102.2 | 70.00 |
| Protein | | 67.97 | 102.2 | 70.57 |
| Ligands | | – | – | 42.92 (NAG) 46.01 (PTY) |
| R.m.s. deviations | | | | |
| Bond lengths (Å) | | 0.004 | 0.004 | 0.004 |
| Bond angles (°) | | 1.03 | 1.06 | 1.01 |
| Validation | | | | |
| MolProbity score | | 1.6 | 1.6 | 1.7 |
| Clashscore | | 7.3 | 10.7 | 7.7 |
| Poor rotamers (%) | | 0 | 0 | 0 |
| Ramachandran plot | | | | |
| Favored (%) | | 96.97 | 97.48 | 95.33 |
| Allowed (%) | | 3.03 | 2.52 | 4.67 |
| Disallowed (%) | | 0 | 0 | 0 |

adopts a clear α-helical conformation in PANX3 and PANX2, instead of a loop observed in PANX1$_{WT}$, constricting the vestibule in the region, thus allowing PANX3 to have an additional vestibule beneath the pore (Fig. 2a, d). The residues F58-S59-S60 line the second constriction point facing the pore in PANX3 and are part of the PANX3 sequence that is variable between PANX1, 2 and 3 (Supplementary Fig. 6d). In contrast, I58 residue in PANX1$_{WT}$ participates in hydrophobic interactions between TMs 1 and 2. Despite sequence variation between PANX2 and PANX3, the F74 in PANX2 forms a similar motif that can allow the demarcation of the vestibule into two regions even in PANX2 isoform (Fig. 2e). The diameter at this constriction point is 21 Å in PANX3 compared to 30 Å in PANX1$_{WT}$ and demarcates the boundary between the anionic surface of the upper compartment compared to the amphiphilic lower compartment in PANX3 (Fig. 1d). Additionally, an annulus of seven uncharacterized densities is observed at this second constriction point in PANX3 that might further contribute to restricting access at this region (Supplementary Fig. 6g).

Given the role of PANX 1 and 3 isoforms in ATP release activity[9,30], we analyzed ATP interactions with the channel through binding analysis of ATP-γS with PANX isoforms utilizing microscale thermophoresis (MST). Both ATP and ATP-γS displayed similar affinities when tested using PANX1$_{WT}$ that encouraged us to use the non-hydrolysable analog of ATP for MST binding assays (Supplementary Fig. 7a, b) The affinity of ATP analog ATP-γS for PANX3 was determined to be 69 μM compared to the 18 μM affinity observed for PANX1$_{WT}$ (Supplementary Fig. 7c). The ATP-γS binding studies suggest a weaker affinity of PANX3 of ATP-γS compared to PANX1. Although we observe a lower affinity of ATP-γS for PANX3, it is difficult to speculate the factors affecting the affinity, as a clear ATP binding site is yet to be detected in large-pore ATP channels such as CALHM, connexins and pannexins.

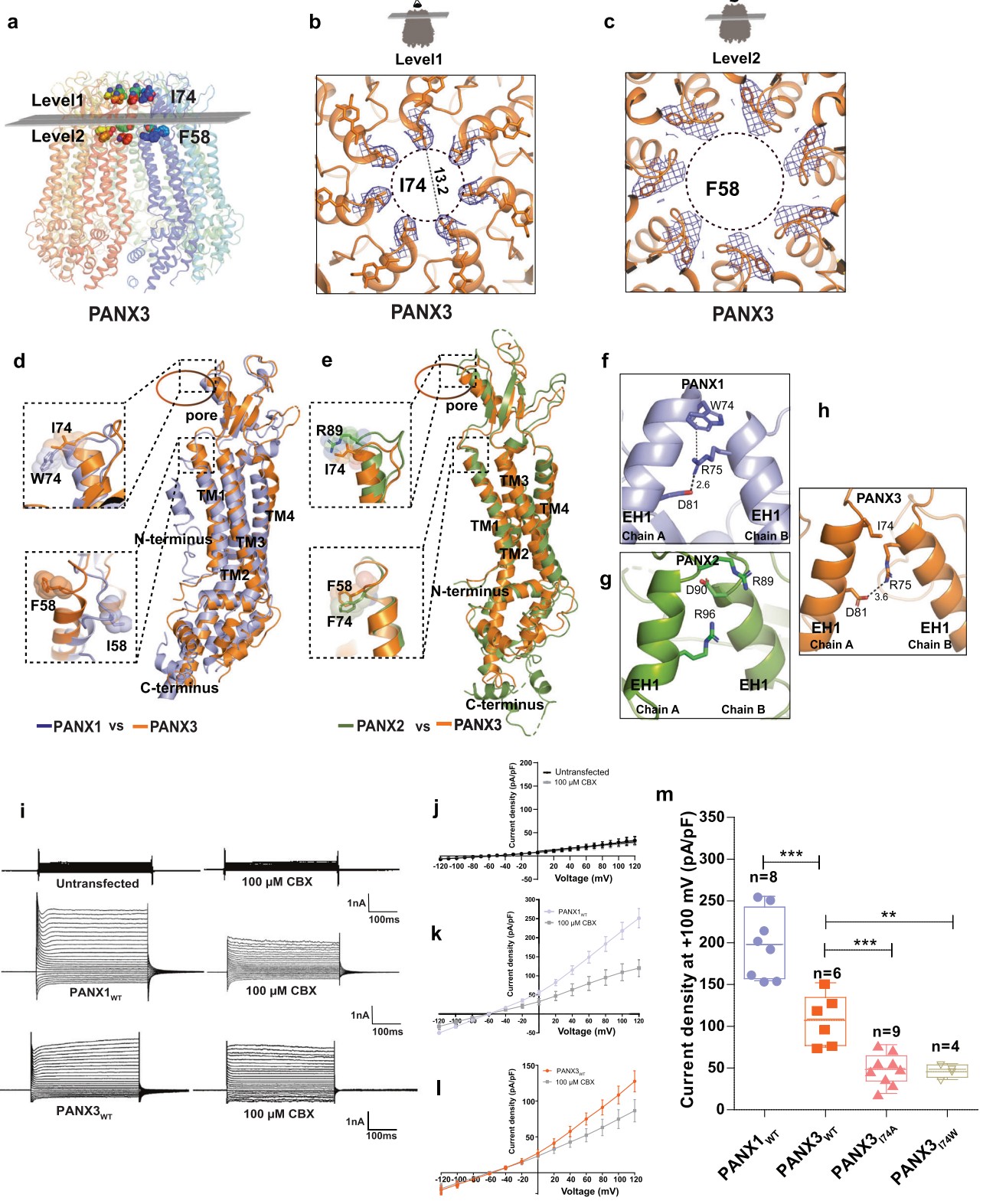

Furthermore, different pore-lining residues observed in PANX1 and PANX3 may dictate isoform-specific differences in channel properties as observed in connexins[35]. Despite earlier unsuccessful attempts to record currents from PANX3[24,25], we could observe currents in PANX3 heterologously expressed in HEK293 cells (Fig. 2i). Whole-cell current elicited by PANX3 shows similar I-V profile as PANX1 with outwardly rectifying currents at positive potential (Fig. 2i-l). We observe weaker CBX sensitivity in PANX3, in comparison to PANX1

(Fig. 2k, l, Supplementary Fig. 7d). The ECD region, particularly loop1, surrounding the pore was previously implicated as being important for CBX interactions[25]. The substitutions among pore lining residues observed in PANX3 could be a reason for the reduced sensitivity to CBX (Supplementary Fig. 7d). This also reinforces the observation of PANX1 pore substitutions, W74I and W74A, that lack sensitivity to CBX inhibition[25]. The ability of CBX to interact with PANX3, albeit weakly, despite a wider pore, strengthens the idea that CBX interactions at

**Fig. 2 | PANX3 pore and affect of substitutions on channel properties.**
**a** Transverse sections of PANX3 at two distinct levels are presented, illustrating the locations of two constrictions. Specifically, I74 and F58 in PANX3 contribute to the formation of the first and second constrictions, respectively. **b** A close-up of the first constriction seen from above in PANX3 formed by I74. The density for the residue (I74) at 7.5 σ is shown. **c** A close up of the second constriction formed by F58 in PANX3. The density for the residue (F58) at 7.5 σ is shown. **d** Superposition of PANX1 and PANX3 displaying the position of residues 58 and 74 with a rmsd of 3.2 Å for 302 Cα atoms, **e** Superposition of PANX2 (PDB ID: 7XLB) and PANX3 exhibits the differences in the pore residues, R89 and I74 in PANX2 and PANX3 respectively with a rmsd of 5.4 Å for 288 atoms aligned; F74 in PANX2 acquires similar position as F58 in PANX3. **f** Cation-pi interaction(dotted lines) between W74 and R75 in PANX1 (PDB ID: 6WBF) is lost in PANX3. **g** Hydrogen bond interaction between R75 and D81 in PANX1 is also observed in PANX3 similar to PANX1. All the distances depicted in the figure are in angstroms (Å). **h** Pore residue (R89, D90) in PANX2 do not form any interactions with neighboring residues unlike PANX1 and PANX3 **i–l**, Representative traces for whole-cell current for HEK293 untransfected cells(mock) and HEK293 cells expressing PANX1$_{WT}$ and PANX3 with and without CBX (100 μM) application.

Current density-voltage plot for untransfected, PANX1 and PANX3 in presence and absence of CBX. Each point represents the mean of $n = 5$ (untransfected), $n = 4$ (PANX1) and $n = 6$ (PANX3) individual recordings, and the error bar represents SEM. **m** Current density is plotted for the PANX1$_{WT}$ ($n = 8$) and PANX3$_{WT}$ ($n = 6$) and its mutants, PANX3$_{I74A}$ ($n = 9$), PANX3$_{I74W}$ ($n = 4$), the error bar represents SEM. $n$ represents the number of cells used for independent recordings; a two-tailed unpaired t-test is used for calculating the significance, ***$p < 0.001$; n.s., not significant, PANX1$_{WT}$ vs PANX3$_{WT}$ ($P$ value < 0.0001), PANX3$_{WT}$ vs PANX$_{I74A}$ ($P$ value = 0.0005), PANX3$_{WT}$ vs PANX$_{I74W}$ ($P$ value = 0.0047), The whiskers represent minimum and maximum value, the left edge of the box represents 25% quartile and the right edge represents 75% quartile, the middle line represents median. Box plot statistics for PANX1$_{WT}$ are, minimum (154.5), 25% percentile (156.9), median (198.0), 75% percentile (242.8), maximum (255.6), for PANX3$_{WT}$, minimum (75.03), 25% percentile (77.00), median (108.6), 75% percentile (134.4), maximum (152.0), for PANX3$_{I74A}$, minimum (49.96), 25% percentile (56.77), median (64.21), 75% percentile (77.97), maximum (97.97), for PANX3$_{I74W}$, minimum (36.24), 25% percentile (38.92), median (48.44), 75% percentile (53.84), maximum (55.14), raw traces and the IV curve for the PANX3 mutants are presented in Supplementary Fig. 12.

extracellular domain can modulate channel activity instead of directly acting as an asymmetric pore blocker. Alternate uncharacterized sites for CBX interactions can exist within PANX isoforms given its sterol-like chemical structure that modulates channel activity. This is reinforced by the lack of saturation in CBX interactions when tested using binding experiments (Supplementary Fig. 7e, f). The structural and the vestibule surface electrostatic differences in the pores observed between the three isoforms PANX1, 2 and 3 can therefore influence their respective channel activities and, consequently, their physiological effects (Fig. 1d). A notable difference between PANX1 and PANX3 is the pore-lining residue, I74 in PANX3 as opposed to W74 in PANX1. To unravel the consequences of this difference, we substituted I74 to a tryptophan (I74W) and alanine (I74A), respectively and observed the impact of these alterations on channel function, in PANX3. We observed that substitutions at this site are not tolerated and lead to a loss of any measurable current and a reduced surface expression among PANX3 substitutions (Fig. 2m and Supplementary Fig. 6b). This is unlike PANX1 where W74I substitution retained PANX1$_{WT}$ like currents[25].

**PANX1 pore substitutions mimic pore residues of PANX2**
As observed in the PANX3$_{I74W}$ mutant, a single mutation at the pore led to an inactive channel. Moreover, single substitutions of W74R and R75D in the pore region yielded incorrectly assembled PANX1[26]. In order to observe if a dual substitution of both the residues (W74, R75) by charged amino acids could compensate for this behavior, we created a PANX1 double mutant (PANX1$_{WR/RD}$) by substituting W74 and R75 with arginine and aspartate, respectively (Fig. 3a). Incidentally, these substitutions are similar to the pore residues of PANX2 revealed by multiple sequence alignment and structural comparison with PANX2 (Supplementary Fig. 1 and Fig. 3b, c). Surprisingly, we obtained a minor fraction of well-assembled PANX1$_{WR/RD}$ particles that allowed reconstruction of the PANX1$_{WR/RD}$ structure to a resolution of 3.2 Å (Table 1 and Supplementary Fig. 2b). Given the high resolution of this reconstruction, the densities along the pore and vestibule were clear for modeling residue sidechains (Fig. 3c). Whilst the global structure resembles the PANX1$_{WT}$, the pore diameter was reduced to 9.3 Å compared to 12.1 Å in PANX1$_{WT}$ (Fig. 3b). The presence of R74 side chain facing the pore renders the pore of PANX1$_{WR/RD}$ highly cationic as compared to PANX1$_{WT}$ and the pore diameter resembles PANX2 (Fig. 3d–f)[17]. The main interactions at the pore in PANX1$_{WT}$ are W74-R75 cation-pi interaction within the subunit and R75-D81 interaction between two subunits (Fig. 2f). As these residues were mutated, the interactions are lost in the PANX1$_{WR/RD}$ (Fig. 3c). Interestingly, these interactions are vital to stabilize the heptamer, and the charge reversal mutant R75E does not form stable heptamers[26]. Instead, in

PANX1$_{WR/RD}$, D75 of one protomer interacts with R74 of another protomer to stabilize the structure (Fig. 3a, c). Apart from the alterations in the pore size, there are no substantial differences induced by the pore mutation, and the overall structure still bears resemblance to PANX1.

We analyzed the differences of ATP-γS interactions with PANX1$_{WR/RD}$ and PANX1$_{WT}$ via MST binding. This revealed a reduced $K_d$ value of 67 μM in PANX1$_{WR/RD}$ compared to ATP-γS binding affinity for PANX1$_{WT}$ (18 μM), suggesting that ATP interactions are moderately affected by these pore substitutions (Fig. 3g and Supplementary Fig. 7b).

We also investigated the physiological consequences of these modifications on the channel behavior. The data revealed a two-fold reduction in current density in PANX1$_{WR/RD}$ compared to PANX1$_{WT}$ but retained an active channel despite the constriction of the pore radius (Fig. 3b, d–f and Supplementary Fig. 8a). In addition, we observed a comparatively higher surface expression for the mutant than PANX1$_{WT}$ (Supplementary Fig. 8b). We also observed a decreased inhibition of PANX1$_{WR/RD}$ currents with CBX compared to PANX1$_{WT}$ since the primary binding site for CBX, W74, was substituted to an arginine (Supplementary Fig. 7d). It is rather interesting that CBX retains minimal interactions with the PANX1$_{WR/RD}$ despite major substitutions in the pore. This further indicates the presence of alternate sites for CBX binding that could modulate channel gating. Additionally, the conductance density shows a fall in channel conductivity for PANX1$_{WR/RD}$, although normalized G-V curve plotted for the double mutant displays an unaltered voltage sensitivity (Supplementary Fig. 8c and Fig. 4e–g). From this, we can infer that the dual pore substitutions done in PANX1 do not influence the voltage-dependent conductance property of the channel.

Our structural findings suggest that the pore radii are inherently linked to the specific residues lining this constriction. This correlation emphasizes the critical role of these residues in determining the dimensions and charge of the pore and, consequently, the functional properties of the PANX channels. It is evident from the functional studies of the three PANX isoforms, substitution of residues in the pore to resemble other isoforms helps alter the channel properties and the ability of inhibitors like CBX to interact with the modified pannexins[18,25,26].

**Vestibular cationic residues in PANX1 alter ATP interactions and channel properties**
In addition to the pore substitutions among PANX isoforms, their ability to release ATP hinges on their capacity to facilitate ATP permeation through the channel prior to being released through the pore and it is likely to interact at different sites as it permeates through the channel. Given the distinct electrostatic surface properties within the

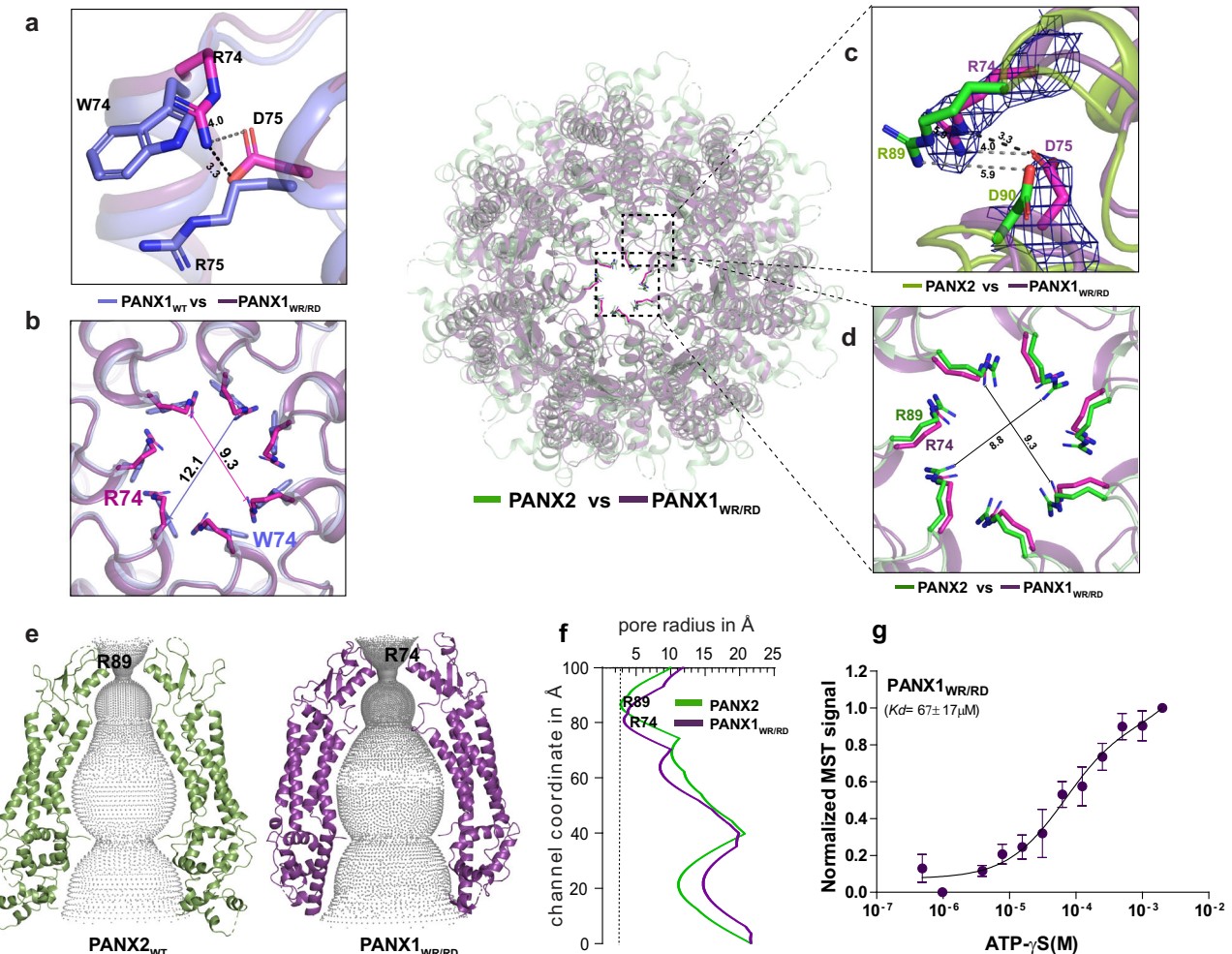

**Fig. 3 | PANX1 pore substitution mutant, PANX1$_{WR/RD}$ constricts the channel.**
**a** The superposition of PANX1 (blue) and PANX1$_{WR/RD}$ (violet) illustrates that in
PANX1$_{WR/RD}$, R74 and D75 serve as pore residues, contrasting with W74 and R75 in
PANX1$_{WT}$. **b** A cross-section of superposed PANX1$_{WT}$ and PANX1$_{WR/RD}$ displays the
reduction of pore. **c** The superposition of PANX1$_{WR/RD}$ and PANX2 unveils the pore
residues in PANX2, with the density for R74 and D75 in PANX1$_{WR/RD}$ contoured at
9.0 σ. **d** A cross-section of superposed PANX2 and PANX1$_{WR/RD}$ displays the position
of arginine at the pore. The distances depicted in the figure are in angstroms (Å).
**e**, **f** The hole profile displays similar pore radius for PANX1$_{WR/RD}$ and PANX2, The
line represents the minimum radius for PANX2 at 2.7 Å formed by the first con-
striction (R89) in PANX2 channel. Units of both X and Y-axes are in angstroms (Å).
**g** The binding affinity for the PANX1$_{WR/RD}$ was determined to be 67 ± 17 μM, $n$ = 3
independent experiments, error bar represents S.D.

vestibule of pannexins, we explored the effects of substituting posi-
tively charged residues along the vestibule in PANX1, like K24, R29 (N-
terminus), K36 (TM1), R75 (EH1, pore), and R128 (TM2), that face the
vestibule of the PANX1 channel (Fig. 4a). With the exception of R128,
these vestibular residues are conserved in both PANX1 and PANX3
isoforms (Supplementary Fig. 1). Through alanine mutations, we sys-
tematically investigated the effects of these substitutions in PANX1 to
decipher their impact on the channel function. The substitution of
R29A and K36A resulted in excessive cell death and consequently, the
purification of these PANX1 mutants was not feasible. The ATP-γS
interaction with K24A substitution has a five-fold loss of affinity ($K$d of
94 μM) compared to PANX1$_{WT}$ (Fig. 4b and Supplementary Fig. 7b). On
the other hand, R128A (TM2) substitution reveals a complete loss of
ATP-γS binding, indicating the importance of this region for PANX1
function (Fig. 4c). Moreover, R75 was suggested as a putative ATP
binding site through alanine scanning mutagenesis[36]. We, therefore,
mutated R75 to alanine and checked the binding affinity of PANX1$_{R75A}$
with ATP-γS. The MST data suggests a decrease in the binding affinity
with PANX1$_{WT}$ but does not abolish it completely (Fig. 4d). Although
we observe that R75 contributes to ATP-γS binding, it is not the sole

residue for ATP binding as substitution of other positively charged
residues also led to decreased ATP-γS binding (Fig. 4b–d). In all like-
lihood, multiple residues are involved in ATP interactions, and
mutating these residues individually reduces ATP binding but does not
abolish it entirely.

To further investigate the effects of these mutations, we per-
formed whole-cell patch clamp studies. The PANX1$_{R128A}$ behaved
similar to the untransfected cells and the loss of ATP-γS binding in
PANX1$_{R128A}$ could be a result of improper oligomerization, as observed
through the widened size-exclusion chromatography profiles (Sup-
plementary Fig. 9). The N-terminus mutant K24A, exhibited weaker
current density compared to the wild type in electrophysiology stu-
dies, which could be linked to its lower surface expression compared
to PANX1$_{WT}$ (Fig. 4e and Supplementary Fig. 8b).

Moreover, electrophysiology data shows a two-fold decrease in
the current density in PANX1$_{R75A}$ compared to PANX1$_{WT}$, although
surface expression for the mutant is significantly higher than the
PANX1$_{WT}$. Since the residue is involved in stabilizing interactions with
W74 and interprotomeric interactions, R75A substitution could locally
alter the interactions around the pore (Fig. 2f). Further, normalized G-V

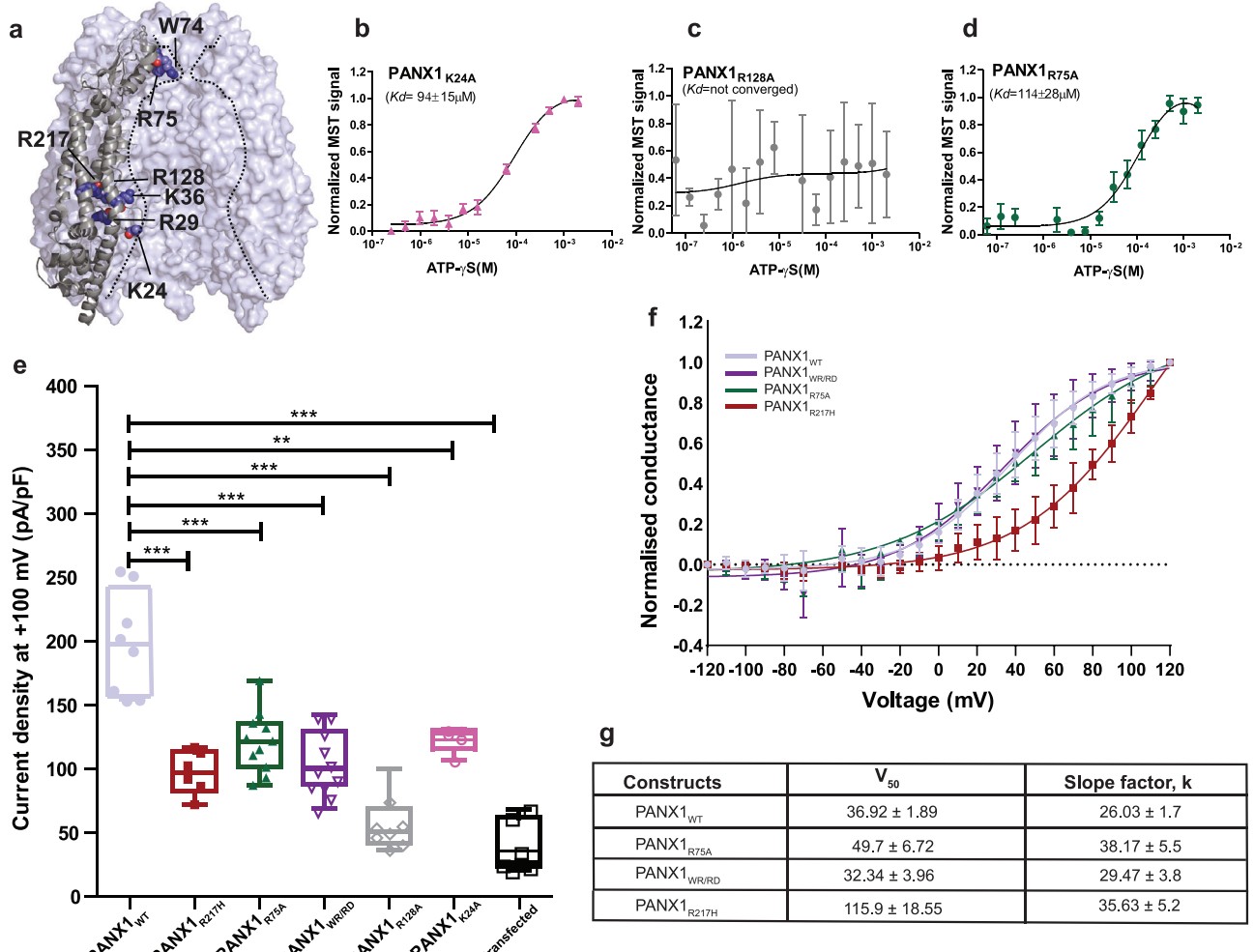

**Fig. 4 | Vestibular cationic residues in PANX1 alter ATP interactions and affect channel currents. a** The position of the mutants selected for ATP-γS binding and whole patch clamp experiments is displayed. **b** Microscale thermophoresis profile for PANX1$_{R24A}$ shows the binding of $94 \pm 15$ μM. **c** MST data reveals a complete loss of binding in PANX1$_{R128A}$ mutant. **d** The binding affinity for the putative ATP binding site (R75) in PANX1 was determined to be $114 \pm 28$ μM, suggesting that R75 is not the sole residue responsible for ATP-γS binding in PANX1, $n = 3$ independent experiments, error bar represents S.D. **e** Current density is plotted for the PANX1$_{WT}$($n = 8$) and the mutants, PANX1$_{R75A}$($n = 11$), PANX1$_{R217H}$($n = 6$), PANX1$_{WR/RD}$($n = 11$), PANX1$_{K24A}$($n = 5$) PANX1$_{R128A}$($n = 7$), and untransfected($n = 7$) the error bar represents SEM. $n$ represents the number of cells used for independent recordings; a two-tailed unpaired $t$-test is used for calculating the significance, ***$p < 0.001$; n.s., not significant, PANX1$_{WT}$ vs PANX1$_{R75A}$ ($P$ value $< 0.0001$), PANX1$_{WT}$ vs PANX1$_{WR/RD}$ ($P$ value $< 0.0001$), PANX1$_{WT}$ vs PANX1$_{R217H}$ ($P$ value $< 0.0001$), PANX1$_{WT}$ vs Untransfected ($P$ value $< 0.0001$), PANX1$_{WT}$ vs PANX$_{K24A}$ ($P$ value $= 0.0022$), PANX1$_{WT}$ vs PANX1$_{R128A}$ ($P$ value $< 0.0001$). The whiskers represent minimum and maximum value, the left edge of the box represent 25% quartile and the right edge represents 75% quartile, the middle line represents median. Box plot statistics are follows, for PANX1$_{WT}$, minimum (154.5), 25% percentile (156.9), median (198.0), 75%

percentile (242.8), maximum (255.6), for PANX1$_{R217H}$ are as follows, minimum (71.99), 25% percentile (82.77.), median (97.37), 75% percentile (113.5), maximum (116.9), for PANX1$_{R75A}$, minimum (87.15), 25% percentile (101.4), median (121.7), 75% percentile (135.7), maximum (168.8), for PANX1$_{WR/RD}$ are as follows, minimum (69.0), 25% percentile (88.1), median (100.2), 75% percentile (129.4), maximum (142.5), for PANX1$_{R128A}$, minimum (36.5), 25% percentile (40.1), median (49.3), 75% percentile (54.9), maximum (73.7), for PANX1$_{K24A}$, minimum (107.0), 25% percentile (115.6), median (127.8), 75% percentile (130.7), maximum (130.9), for untransfected, minimum (20.9), 25% percentile (23.5), median (27.0), 75% percentile (62.0), and maximum (68.6), raw traces and the IV curve for the PANX1 mutants are presented in Supplementary Fig. 11. **f** Normalized conductance-voltage (GV-curve) plot for PANX1$_{WT}$($n = 8$), PANX1$_{R75A}$($n = 6$), and PANX1$_{WR/RD}$($n = 4$) suggests that the pore residues are not involved in the voltage sensitivity of the channel, conductance-voltage(GV-curve) plot for PANX1$_{R217H}$ exhibits a reduction in the voltage sensitivity of the channel, $n = 4$; the error bar represents SEM. **g** The normalized G-V values were fitted with the Boltzmann equation, and the voltage at which the half-maximal activation, V$_{50}$, occurred along with slope factor, $k$, was calculated for all the constructs.

curve for R75A is comparable to wild type and reveals that the substitution of a pore residue, R75, although alters the channel properties but does not affect the voltage sensitivity of the channel (Fig. 4f, g) consistent with the findings with ZfPANX1[37]. Similar consequences to arginine substitutions in the pore are observed with LRRC8A, where the channel maintains conductance with arginine substitution but loses anion selectivity[38].

It is evident from these observations that the charge substitutions in the vestibule and the pore can influence ATP interactions with the PANX1 channel (Fig. 4b–d and Supplementary Fig. 8a–c).

**Pannexin 1 germline mutant R217H leads to pore constriction**

Encouraged by structural and functional alterations observed in both pore and vestibular mutants of PANX1, we explored the impact of a transmembrane germline mutation on the inherent properties of the channel. We elucidated the structure of PANX1$_{R217H}$ (TM3) substitution to characterize the effects of this germline mutant and to understand the basis of its defective channel properties. The arginine residue at the 217 position (TM3) is highly conserved among vertebrate PANX1 orthologues and is buried amongst TM helices and does not display solvent access unlike the substitutions studied earlier

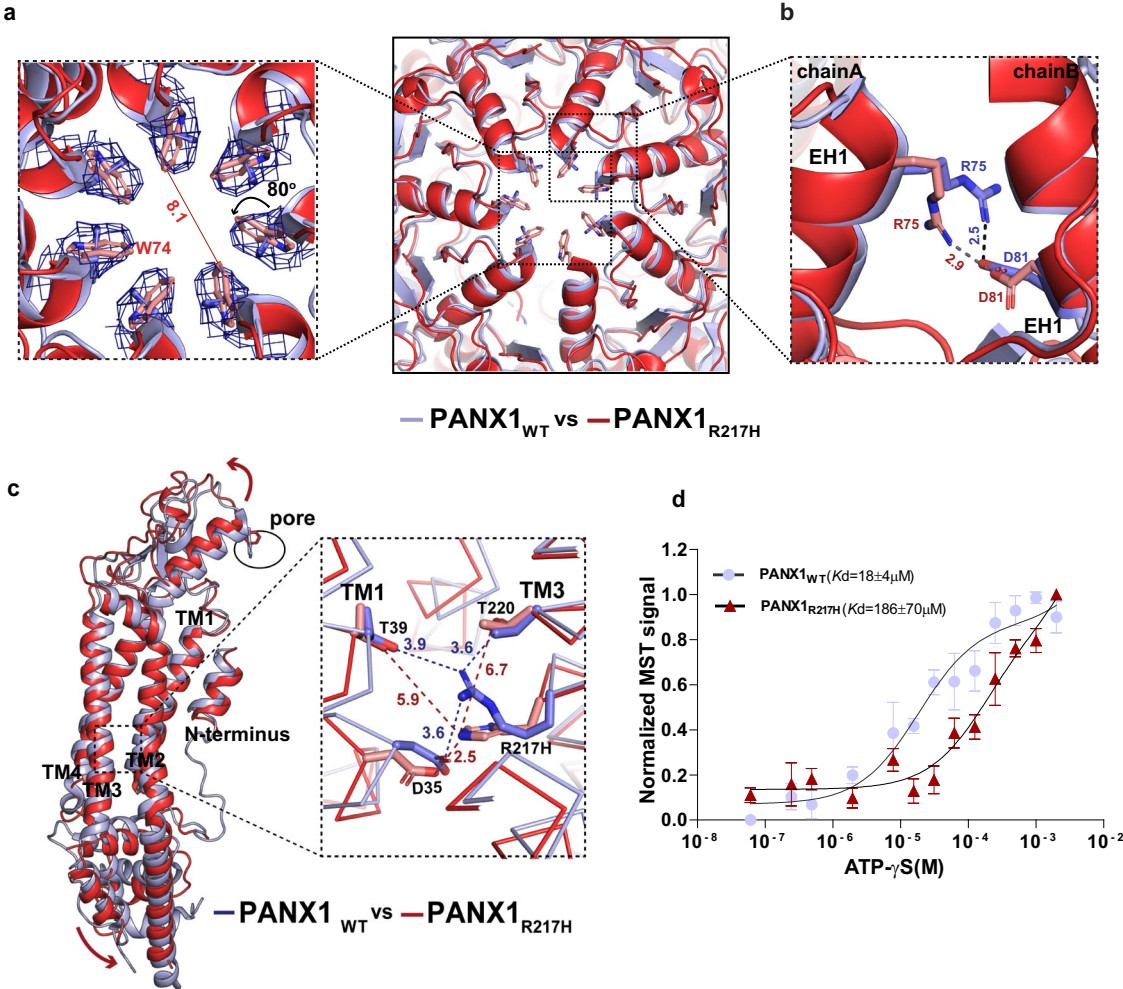

**Fig. 5 | PANX1 germline mutant, PANX1$_{R217H}$ alters channel properties. a** A cross-section of superposed structures of PANX1$_{WT}$ (blue) and PANX1$_{R217H}$ (red); the inset shows the altered conformation of the residue (W74) at the extracellular entrance of the pore. The dashed line represents the reduced pore cross-section distance in PANX1$_{R217H}$ (red); the W74 rotates by 80° at the χ2 torsion angle, reducing the pore diameter by 3.8 Å compared to PANX1$_{WT}$ (Supplementary Movie 1). The density corresponding to W74, depicted at 5.0 σ, is illustrated in the PANX1$_{R217H}$ mutant channel. **b** The residue R75 in PANX1$_{R217H}$ forms a salt bridge interaction with D81 of the adjacent subunit, mirroring the interaction observed in PANX1$_{WT}$.
**c** The structural superposition of PANX1$_{WT}$ and PANX1$_{R217H}$ exhibits a disrupted hydrogen-bond network, owing to the mutation, displayed in the inset. For clarity, only one subunit is shown, and arrows indicate the direction of the movement of the mutant in comparison to PANX1$_{WT}$. The distances displayed in the figure are in angstroms (Å). **d** Weak apparent binding affinity of the PANX1$_{R217H}$ with ATP-γs was determined to be 186 ± 70 μM compared to 18 ± 4 μM of PANX1$_{WT}$, $n$ = 3 independent experiments, error bar represents S.D.

(Fig. 4a and Supplementary Fig. 10a). Consistent with the earlier reports[15], we observed comparable expression of the PANX1$_{R217H}$ channel to PANX1$_{WT}$ (Supplementary Fig. 10b). The structure was determined by cryo-EM to a resolution of 3.9 Å (Supplementary Fig. 2c and Table 1). We further improved the resolution of ECD and TMD domains to 3.77 Å through focussed refinement (Supplementary Fig. 10c, d). However, we were unsuccessful in improving the resolution of ICD domain through this step. The superposition of PANX1$_{WT}$ and PANX1$_{R217H}$ mutant yielded a rmsd of 1.6 Å for 288 Cα atoms.

Despite the lower resolution of the structure, it resembles PANX1$_{WT}$ globally, with subtle changes in extracellular domain (ECD), extracellular helix 1 (EH1), and the intracellular domain (ICD). Extracellular loops and EH1 have an average shift of 1.4–1.6 Å away from the pore. The W74 residue of PANX1$_{R217H}$ fits better into the density of its sidechain with a rotameric shift that coincidentally constricts the pore radius (Fig. 5a and Supplementary Fig. 10c). Despite a shift in the side chain position of R75, this residue retains interactions with the D81 of the adjacent subunit through a salt bridge, consistent with PANX1$_{WT}$ (Fig. 5b).

We investigated the residue environment in the vicinity of R217 within a 4 Å radius in TM3 and observed hydrogen bond interaction with D35 in TM1 (3.6 Å) in PANX1$_{WT}$. R217 also interacts with T220 in TM3 and T39 in TM1 through hydrogen bonds (Fig. 5c). Mutating arginine to histidine disrupts the H-bond interaction network within TMs 1 and 3. The H-bond interaction of T220 in TM3 and T39 in TM1 with R217H is disrupted due to the shorter side chain of histidine compared to arginine. The G44 in TM1 that likely acts as a hinge point facilitates the TM1 bending and displacement as a consequence of R217H substitution. The displacement observed in TM1 translates to an outward movement of ECD that could facilitate greater flexibility of the pore lining residue W74 to constrict the channel (Fig. 5a).

In comparison to the PANX1$_{WT}$, the modeled side chain W74 has a χ2 torsion angle shift of nearly 80° towards the pore leading to a constriction of the pore diameter in the mutant PANX1 (Fig. 5a, c). The current density for PANX1$_{WT}$ was twofold higher than PANX1$_{R217H}$, at a positive voltage of 100 mV which could be associated with the partial closure of the pore (Fig. 4e, Supplementary Fig. 8a). Although there was a significant decrease in the current density in the R217H mutant,

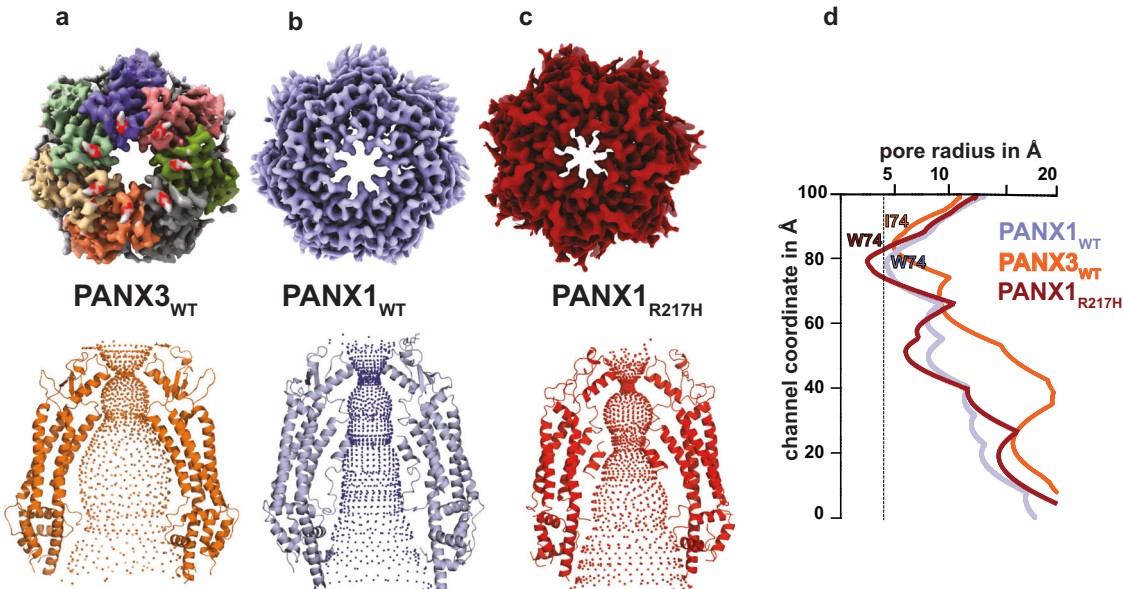

**Fig. 6 | Representation of ion/ATP conducting pathway in PANX1 and PANX3.**
The minimum radius is observed at the first constriction in PANX isoforms. **a** The minimum radius is 4.8 Å for PANX3 at the constriction formed by I74. **b** The PANX1$_{WT}$ has a minimum radius of 4.2 Å formed by W74. **c** The constriction point is created by W74 in PANX1$_{R217H}$. The constriction point is 2.4 Å in PANX1$_{R217H}$. **d** 2D representation of the hole profile. The line represents the minimum radius for PANX1$_{WT}$ at 4.2 Å formed by the first constriction (W74) in PANX1 channels.

the mutation did not have significant effect on the CBX binding, as we could observe the inhibition of PANX1$_{R217H}$ currents by CBX similar to PANX1$_{WT}$ (Supplementary Fig. 7d). A comparison of the conductance density (nS/pF) to voltage reveals highly weakened channel conductance in response to increasing voltage in comparison to PANX1$_{WT}$ (Supplementary Fig. 8c). A comparison of normalized conductance to voltage for PANX1$_{R217H}$ and PANX1$_{WT}$ reveals altered voltage sensitivity ($V_{50}$) of the mutant channel from around 40 mV for the PANX1$_{WT}$ to over 100 mV indicating reduced voltage sensitivity as a consequence of R217H mutation (Fig. 4f, g).

Binding studies using MST display a significant decrease in ATP-γS binding in the PANX1$_{R217H}$ mutant compared to PANX1$_{WT}$ suggesting that mutant behaves differently than the PANX1$_{WT}$ and have reduced ability to bind to ATP-γS (Fig. 5d). Although the R217H substitution is in the TM3, we detect structural shifts in the ECD indicating allosteric effects of this germline mutant. Such allosteric effects are observed in the case of disease-causing mutants where the mutation site is far from the observed structural changes and affects their functional properties[39]. It was proposed in an earlier study that the PANX1$_{R217H}$ interactions with the C-terminus can cause the altered channel properties in this germline mutant[40]. Given the allosteric effects of R217H on the pore diameter observed in this study, long-range effects in C-terminus could also drive some of the properties observed with PANX1$_{R217H}$. However, given the absence of a structured C-terminus in PANX1 structures determined thus far, the influence of the C-terminus on PANX1$_{R217H}$ structure would be speculative at this juncture.

## Discussion

In this study, we conclude three major structure-function aspects among PANX isoforms. Firstly, we report the structure of human PANX3, an isoform of PANX1 that allows a comprehensive structural comparison among pannexin isoforms. PANX3 displays a heptameric oligomer assembly consistent with PANX1 and PANX2[19]. We could observe outward rectifying currents similar to PANX1 as it responds to positive voltage but has a lower affinity for ATP-γS and is observed with the widest pore of 13.2 Å among the three isoforms suggesting that

PANX3 is in a conformation with an open pore (Fig. 6a). A motif in TM1 facilitates the formation of a second anionic vestibule in the neck region around F58 in PANX3 structure. The flexibility at the side chain could dynamically regulate the size of this constriction point. This could aid in the regulation of PANX3 channel activity, particularly in the context of PANX3 lacking a long C-terminus that is caspase-sensitive, unlike PANX1. The presence of hydrophobic residues at the neck has been observed in ion channels like Bestrophin[41] and pentameric ligand-gated ion channels (pLGICs)[42]. Hydrophobic constrictions in ion channels can aid in the creation of barriers to ion flow and alter the wettability of the pore[43].

Secondly, the electrostatics of the vestibules within PANX isoforms differ substantially and could influence the permeation of ATP. The charge distribution within the vestibules of ion channels tend to influence the permeation of ions[44]. The observations from the binding assays with an ATP analog and substitutions at multiple vestibular cationic residues reflect this effect wherein individual substitutions along the vestibule seem to affect the binding affinity of ATP and are likely to influence the ability of the channel vestibule to sequentially bind and unbind ATP resulting in its permeation and release. Performing these substitutions close to the pore also influences the activity of the channel as observed in the case of a substitution at R75 which is next to the pore lining W74 in PANX1[26]. Surprisingly, in PANX1, we could attain a reasonably stable pore architecture by having a double substitution of residues in PANX1 pore, W74R/R75D (PANX1$_{WR/RD}$) that also exhibits a smaller, highly cationic pore than PANX1$_{WT}$, consistent with PANX2. The PANX1$_{WR/RD}$ pore resembles LRRC8A, where the constriction is lined by a ring of positively charged arginine residues and plays an essential role in anion permeability[38]. In other isoforms of LRRC8, the arginine is replaced by leucine and phenylalanine akin to what we observe with PANX3 that has an isoleucine (I74) around its pore.

Thirdly, the substitution of a charged residue in TM3 (R217H) to create a germline mutant, PANX1$_{R217H}$, resulted interestingly, in a narrower pore in comparison with PANX1$_{WT}$ (6WBF) as a consequence of the rotation of W74 at its χ2-torsion angle (Supplementary Movie 1). Although reported at a lower resolution compared to other structures

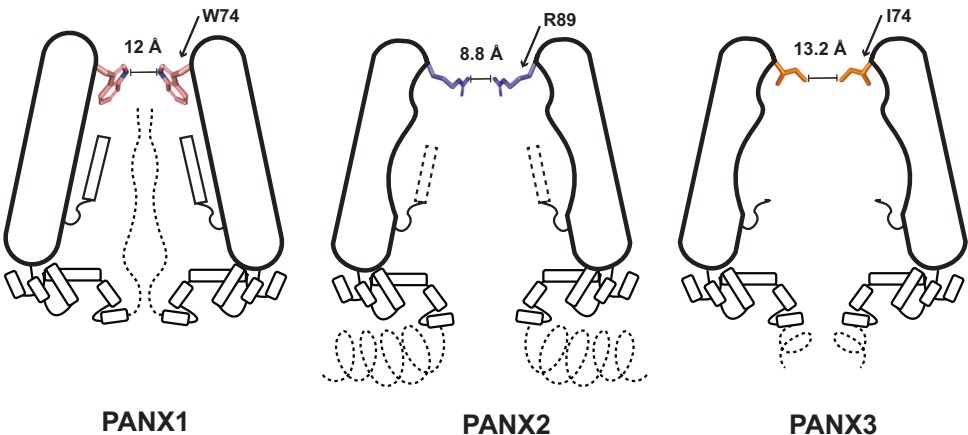

**Fig. 7 | Schematic of PANX isoforms.** A schematic of the three PANX isoforms displaying differences in their structural organization. The residues lining the pore at the extracellular domain are distinct among the three isoforms. The distance between residues at the constriction point are indicated.

of PANX1, the mutant PANX1 displays subtle long-range structural shifts that affect channel function by causing a constriction of pore radius. The weakened conductance behavior of this mutant signifies its inability to respond to a voltage similar to PANX1$_{WT}$. The transition in pore diameter observed in this study as a consequence of a TM substitution indicates allostery between the transmembrane helices and the extracellular domain of PANX1 that also correlates with the reduced ATP efflux observed with this germline mutant of PANX1[15].

The study therefore provides a holistic comparison of the architecture of the three PANX isoforms with an in-depth analysis of the pore radii and the residues that surround it. The differences in the channel pore of the PANX3 and PANX1 mutants are compared to PANX1$_{WT}$[45] wherein the main distinction lies in the width of the pore entrance depicting mutants' smaller size than PANX1$_{WT}$. The smallest Van der Waal radii found were 2.4, 4.8, and 4.2 Å for PANX1$_{R217H}$, PANX3, and PANX1$_{WT}$, respectively, at the pore entrance (Figs. 6a–d and 7). The changes to the pore radii provide functional insights into the behavior of these large-pore ion channels that impact diverse aspects of cellular physiology and disease.

## Methods

### Plasmids and cloning
Full-length human PANX1 (UniprotID-Q96RD7) and PANX3 (UniprotID-Q96QZ0) were synthesized by Geneart (Invitrogen). The synthesized genes were subcloned into a pEG-BacMam vector between EcoR1 and Not1 restriction sites with a C-terminal TEV protease site (ENLYFQ) followed by enhanced green fluorescent protein(eGFP) and 8x His-tag[46].

The site-directed mutagenesis was used to generate the mutants by the mega primer-based whole plasmid amplification method. The primers were designed using SnapGene and synthesized by Sigma (Supplementary Table. 1). The constructs and substitutions in the study were verified by Sanger sequencing.

### Transfections and fluorescence-detection size exclusion chromatography
HEK293S GnTI⁻ cells were seeded at a density of $1 \times 10^6$ cells in DMEM medium with 10% FBS. The cells were transfected using lipofectamine 3000 (Invitrogen) according to the manufacturer's protocol. For fluorescence-detection size exclusion chromatography (FSEC), cells were harvested after 36 hours and were solubilized using 200 µl of 25 mM Tris pH 8.0, 100 mM KCl, 1% glycerol, and 10 mM glycodiosgenin (GDN). The solubilized cells were spun down at 66,000×g for 1 h, and the supernatant was loaded onto the Superose6 increase 10/300 GL column (GE Healthcare). GFP fluorescence ($\lambda_{Ex} = 488$ nm,

$\lambda_{Em} = 509$ nm) was monitored to check the elution volume and the homogeneity of the protein.

### Protein expression and purification
The protein was expressed using the *Bacmam* system for high-level expression[46]. In brief, *E.coli* DH10bac cells were transformed with a pEG-PANX1/3 vector. Positive colonies were selected based on blue-white screening, and bacmids were isolated and transfected into Sf9 adherent cells using cellfectin (Invitrogen) according to the manufacturer's protocol. Four days post-transfection, cells were visualized for GFP fluorescence under a fluorescence microscope (the presence of green cells implies the presence of the virus). The virus was filtered through a 0.22 µm filter and harvested. Sf9 cells in suspension at a cell count of $2 \times 10^6$ cells per ml were infected for the generation of the P2 virus. Four days after the infection, cells were spun down at 7000 g, and the supernatant was filtered and stored at 4 °C, protected from the light.

HEK293S GnTI⁻ cells were grown at 37 °C in Freestyle 293 medium (Invitrogen) supplemented with 1.5% FBS. Cells were infected with the P2 virus at a density of $2.5–3.0 \times 10^6$ cells per ml. Twelve hours post-transfection, sodium butyrate was added at a final concentration of 5 mM, and the temperature was reduced to 32 °C. Cells were harvested 60 h after infection for the enrichment of the membranes, and cells were sonicated for 15 min at 35% amplitude; the sonicated cells were centrifuged at 100,000×g for one hour. Membranes were flash-frozen and kept in −80 °C till further use.

The expressed PANX1/3 was extracted from membranes using 10 mM glycodiosgenin (GDN), 25 mM Tris pH (8.0), and 100 mM KCl. The solubilized membranes were centrifuged at 100,000 × g for one hour. The supernatant was incubated with Nickel-NTA resin (Qiagen) equilibrated with binding buffer, for two hours. The protein was eluted in 25 mM Tris pH (8.0), 100 mM KCl, 1% glycerol, 0.3 M imidazole, and 100 µM GDN. The eluted protein was treated with TEV protease for C-terminal GFP and 8x-his-tag removal.

The GFP cleaved protein was further purified by size exclusion chromatography in 25 mM Tris pH (8.0), 100 mM KCl, 1% glycerol, and 50 µM GDN (SEC buffer) using Superose6 increase 10/300 GL column. The peak fraction was collected and used for grid freezing (Supplementary Figs. 9 and 15).

### Cryo-EM sample preparation and data collection
The SEC-purified PANX1/PANX3 was concentrated to 5 mg/ml using a 100 KDa cut-off concentrator (Millipore). The concentrated protein was centrifuged at 66,000×g for one hour before grid freezing. Quantifoil 300 mesh gold holey carbon grids (R1.2/1.3) were glow discharged in the air for one minute at 25 mA, and the protein (3 µl) was

applied to the grid in ThermoFisher vitrobot at 100% humidity and 16°C temperature. The grids were blotted for 3.5 s with a wait time of 10 seconds. For the $PANX1_{R217H}$ mutant, samples were applied twice, and grids were blotted sequentially for 2.0 and 3.5 seconds, respectively. The grids were flash-frozen in liquid ethane and were stored in liquid nitrogen till further use.

For PANX3, ATP at a final concentration of 1 mM was added 30 min prior to the grid freezing. The ATP stock was prepared in SEC buffer (Tris pH 8.0).

Datasets ($PANX1_{WT}$, $PANX1_{R217H}$, PANX3) were collected on a Titan Krios 300 keV (ThermoFisher) equipped with Falcon 3 or K2 direct electron detectors (Table 1). Falcon 3 detector was used For $PANX1_{WT}$ data collection, whereas data for $PANX1_{R217H}$ and PANX3, data was collected on the K2 detector in EFTEM mode with a Bio-quantum energy filter and 20 eV slit width. The pixel size of 1.07 Å was used for both the detectors and the total electron dose for different constructs is mentioned in the table (Table 1). For $PANX1_{WR/RD}$, the data was collected at CM01, European synchrotron and Radiation Facility (ESRF), France, on the K3 detector with an energy filter of 20 eV (Table 1).

### Cryo-EM data processing and model building
PANX1/3 structures were determined using the cryoSPARC version (3.20)[47]. The movies were imported and motion-corrected by patch motion correction, and the contrast transfer function (CTF) was estimated by patch CTF. After manually curating the images, the CTF-estimated micrographs were used for auto-picking by the blob picker. For manual curation, a cut-off of 6 Å for CTF estimation and 1.06 for ice thickness was used to remove bad micrographs.

The auto-picked particles were extracted with a box size of 320/360 pixels and were subjected to 2D classification; two rounds of 2D classifications were done to remove the junk particles. The selected 2D classes were subjected to ab-initio initial model generation and 3D classification. For the ab-initio modeling, the initial and final resolutions were kept at 12 Å and 8 Å, respectively, and the minibatch size of 1000 was used. The high-resolution class was used for non-uniform refinement[48]. The parameters for non-uniform refinement were adjusted to get a higher resolution map. The maximum alignment resolution was changed to the Nyquist limit, the number of extra passes was increased to 2, the initial dynamic mask resolution was changed to 14 Å, and the batch size was kept like that of ab-initio modeling. The map obtained from non-uniform refinement was subjected to local refinement; a separate mask for the local refinement was not generated, and a default mask provided by cryoSPARC was used. PANX1 structure refinement was performed with C7 symmetry, whereas PANX3 was processed with C1 for the initial refinement; as C7 symmetry could be seen in the ab-initio 3D structure, C7 symmetry was applied during the non-uniform refinement to improve the resolution. PANX1 mutants were modeled using previously determined structures (PDB-ID: 6WBF). The desired residues were mutated, and the changes owing to the mutations were modeled in the cryo-EM density map in Coot[49].

For PANX3, an Alphafold2 monomer was fitted in the map to make a heptamer in Chimera. The model and the map were aligned using autodock in Phenix. The model was built in the density in coot. All the structures were refined using PHENIX real-space refinement[50]. The workflow and the statistics for each construct are summarized in the Supplementary Figs. 2 and 3 and Table 1.

### Binding studies with ATP-γs
Binding studies were done using microscale thermophoresis (MST). The protein concentration of 10 nM (calculated for the heptamer) was kept constant for all the studies. The protein was labeled with 10 nM red Tris Pico dye, and Monolith standard treated capillaries were used to detect binding. A non-hydrolyzable ATP analog, ATP-γs, dissolved in SEC buffer, was used as a ligand for the binding studies. The ligand

concentration was diluted (2x) for the study in 16-serial dilution, with 2 mM as the highest concentration. The final protein concentration was kept at 5 nM. All the experiments were done in triplicates.

### Electrophysiology
HEK293 cells were maintained in DMEM F-12 Ham medium (Sigma) supplemented with 10% fetal bovine serum (GIBCO, heat-inactivated US origin) and 1% antibiotic-antimycotic solution (Sigma) in a humidified incubator at 37 °C with 5% $CO_2$. The cells were passaged twice a week, and a fraction of the cells were plated onto 35 mm cell culture dishes (Thermofisher). PANX1, PANX3, and their mutants were transiently transfected into HEK293 cells with an enhanced green fluorescent protein (eGFP) using the transfection agent Lipofectamine 2000 (Invitrogen). The cells expressing GFP were selected for patch clamp electrophysiological recordings after 24–36 h of transfection.

PANX currents were activated and recorded in whole-cell mode using EPC 800 amplifier (HEKA Elektronik), Digidata 1440 A digitizer (Molecular Devies), and pClamp 10 software. To elicit the currents, 400 ms voltage clamp steps were applied from a holding potential of −60 mV to test potentials of −120 to +120 mV in 10/20 mV increments. The currents were sampled at 20 kHz, and digitally low pass (Bessel) filtered at 3 kHz. The patch electrodes used for electrophysiology experiments were fabricated using borosilicate glass capillaries and had a resistance of 3−5 MΩ when filled with an internal solution. The pipette solution contained (in mM) 150 Cesium gluconate, 2 $MgCl_2.6H_2O$, and 10 HEPES (pH adjusted to 7.4 with CsOH). The extracellular solution contained (in mM) 147 NaCl, 2 KCl, 1 $MgCl_2.6H_2O$, 2 $CaCl_2.2H_2O$, 10 HEPES, 13 Glucose (pH adjusted to 7.4 with NaOH)[15]. The effects of carbenoxolone (Sigma) were evaluated at a final bath concentration of 100 μM. The experiments were performed at room temperature (23 °C; Supplementary Figs. 11 and 12).

Electrophysiological data were analyzed using Clampfit, Microsoft Excel, and GraphPad Prism. The current−voltage relationships were plotted as normalized steady-state current values at the end of the pulse step (at 400 ms) versus the respective voltage step (mV). Current densities (pA/pF) were obtained by dividing the steady-state current values(pA) by the cell membrane capacitance (pF). The conductance voltage relationship was plotted to analyze the voltage dependence of channel activation. The conductance (G) was calculated using $I = G(V_t − V_{rev})$, where $I$ is the steady state current value at the test potential $V_t$, and $V_{rev}$ is the reversal potential. The conductance voltage plot was fitted to a Boltzmann equation: $I = I_{max}/1 + exp((V_t - V_h)/k)$ where $I_{max}$ is the maximum steady state current amplitude, $V_h$ is half maximal voltage for activation ($V_{50}$), and $k$ is the slope factor. The number of recordings (n) is mentioned in the figure; a two-tailed unpaired t-test is used for calculating the significance, ***$p < 0.001$; n.s., not significant.

The conductance density was calculated by dividing the conductance values (nS) by the cell membrane capacitance (pF).

### Immunofluorescence staining
The HEK293 cells were transfected with GFP-tagged human PANX1 and 3 isoforms and their respective mutants using Lipofectamine 3000 reagent. After a 24-h transfection period, the cells were fixed with 4% paraformaldehyde for 15 min and subjected to two washes with phosphate-buffered saline (PBS) buffer. Subsequently, the cells were treated with wheat germ agglutinin (WGA) for 10 min at 37 °C, following the manufacturer's instructions.

Post-staining, the cells were mounted using 4′,6-diamidino-2-phenylindole (DAPI) Fluro mount-G after PBS washing and were then analyzed utilizing a Leica SP8 Falcon microscope (Bioimaging Facility, IISc, Bangalore, India). For each construct, more than 40 cells from a minimum of five distinct coverslips were selected for colocalization analysis. WGA was utilized as a marker for the cell membrane, and the expression levels of PANX1 and its isoforms, including the mutants, were quantified based on their GFP intensities.

Specifically, the colocalized GFP intensity was quantified and normalized by the total WGA intensity, providing insight into the membrane expression levels of WT PANX1 and 3 and their mutants. The colocalization ratios of the GFP+ area in the WGA+ area, indicative of protein expression levels on the plasma membrane, were determined using FIJI software. All data were collected and subjected to analysis using GraphPad Prism. (Supplementary Figs. 13 and14).

## Reporting summary

Further information on research design is available in the Nature Portfolio Reporting Summary linked to this article.

## Data availability

The structures have been deposited with the following accession numbers in the PDB. 8GTR (PANX3$_{WT}$ structure), 8GTS (PANX1$_{R217H}$ structure), 8GTT (PANX1$_{WR/RD}$ structure). The Cryo-EM density maps have been deposited in EMDB under accession numbers, EMD-34265 (PANX3), EMD-34268 (PANX1$_{WT}$), EMD-34266 (PANX1$_{R217H}$), and EMD-34267 (PANX1$_{WR/RD}$). Source data is provided along with the manuscript. Source data are provided with this paper.

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

## Acknowledgements

Research in this manuscript is funded by the Ministry of Education, Govt. of India, (MoE-STARs program) grant to AP (MoE/STARS1/167). The work in the project was partly funded by the WellcomeTrust/DBT-India Alliance intermediate (IA/I/15/2/502063) fellowship. A.P. is senior fellow of DBT-Wellcome Trust India Alliance and an EMBO Global Investigator. N.H. is a PhD student funded by the DBT-JRF program (DBT/2017/IISc/877) at IISc. A.A. is a PhD student funded by the IISc-GATE fellowship. S.M. was a graduate student at IISc funded by the DST-INSPIRE fellowship. SP was a postdoctoral fellow of the DBT-RA program in biological sciences. APB is an early-career fellow of the DBT-Wellcome Trust India Alliance. We thank Sucharita Bose and the National Electron Cryo-Microscopy facility, BLiSc, Bangalore (DBT/PR12422/MED/31/287/2014), for the screening of grids and data collection. We acknowledge the DBT-ESRF access program (BT/INF/22/SP22660/2017) and the European Synchrotron Radiation Facility for provision of time on CM01 microscope and we would like to thank Gregory Effantin, Eaazhisai Kandaiah, and Romain Linares for assistance. We would like to acknowledge the advanced cryo-EM facility, IISc funded through the DBT-BUILDER program (BT/INF/22/SP22844/2017) and DST-FIST program (SR/FST/LSII-039/2015) for screening and preliminary data collection. Computational support from the high-performance computing facility "Beagle" setup from grants by a partnership between the DBT, India, and the Indian Institute of Science (DBT-IISc partnership program) is acknowledged. The authors would like to acknowledge the DBT-IISc partnership program phase II and the DST-FIST program support for research. We would like to thank Subbarao Gangisetty and Deepak Nayak for reagents and help with confocal microscopy at the Divisional Bio-imaging facility at IISc.

## Author contributions

N.H. performed the molecular cloning, ATP binding experiments, immunofluorescence staining and analysis, optimized purification, sample preparation, grid freezing, performed data processing, refinement of PANX structures and data analyses. AA performed and analyzed all the electrophysiology experiments under guidance from S.K.S.; S.P. initiated the cloning of PANX1 gene and optimized heterologous expression. S.M. performed initial electrophysiology measurements. A.P.B. aided in the preliminary experiments with PANX3. K.R.V. performed the data collection, initial data processing, and optimization of grid freezing for PANX1. A.P. designed and planned the project and provided inputs for structure refinement and data analyses. N.H. and A.P. wrote the manuscript with inputs from all authors.

## Competing interests

The authors declare no competing interests.
