## [Peer Review File · Nature Communications]

Cryo-EM structures of Pannexin 1 and 3 reveal differences among Pannexin isoformsReviewers' comments:

Reviewer #1 (Remarks to the Author):

The manuscript by Hussain et al. describes the structural comparison between Pannxin1, Pannexin1 mutant, Pannexin2 mimetic Panx1 mutant, and Panx3. Pannexin is the major large-pore channel permeable to small anions (Cl⁻) and ATP. The Panx family comprises three proteins, Panx1-3. Panx1 is the most dominantly expressed protein studied extensively, whereas Panx2 and Panx3 are understudied. Several Panx1 structures were published in 2020, starting with Michalski et al. (eLife 2020). The Panx1 structure showed narrow constrictions at the extracellular region and another semi-narrow constriction in the TMD. In this study, the authors made Panx1 mutant W74R/R75D that mimics Panx2, and the Panx1 R217H mutant. Other groups posted the Panx2 and Panx3 structures in BioRxive but have not published them yet. The major weakness of this manuscript is that the authors do not clearly explain their structures and structure-based mechanisms. For example, why does Panx3 have a lower affinity to ATP? Why do they have lower sensitivity to CBX? They state that the substitution of W74 causes that, but how? The messages are not clear. Overall, this study provides structures but not mechanisms. Furthermore, several citations need to be added appropriately to improve the level of accuracy of this manuscript.

Specific points:

-The title is misleading since this work does not address any dynamics. The work is strictly on the structural comparison.

-While providing the field with Panx2 and 3 structures is important, the authors must state why they worked on it. For example, are there any specific functional features of Panx2 or 3 the authors want to explain? If so, why are they important in the context of general biology? It should be stated in the manuscript.

-What are the functional states of the cryo-EM structures? The authors need to explain. Given that the C-terminus is not cleaved by a caspase, could it represent the inactive state?

-pg3: Citation 16/19. Detergents are mostly bound to what was published as the CBX binding site. For example, the apo-structure of pannexin has the same density around W74.

-pg 6: With respect to the observation that positively charged patches are present, stating that it may be related to ATP binding and release is casual. It has to be experimentally proven.

-pg6: ATP binding assay with MST – 1) it would be necessary to show traces from the actual experiments, especially since the signals in MST experiments vary. 2) The difference of ATP binding on Panx1 and 3. Could the authors address why there are differences in their structures??

-Some citations need to be added or corrected.

1) Pg 3: The Syrjanen and Michalski review (2021 JMB) wraps up the recent progress in the large-pore channel field and should be a useful one to cite.

2) pg5: CALHM citation. Syrjanen et al. (NSMB, 2020) is the first paper to show the diversity in the oligomeric state and the mechanism to control it.

3) Michalski and Syrjanen et al. (eLife 2020) must be cited in the introduction since that represents the first structural paper on Pannexin.

Reviewer #2 (Remarks to the Author):

The manuscript by Hussain et al, "Structural insights into pore dynamics of human Pannexin isoforms" describes a structural and biochemical/biophysical study of two Pannexin ion channels, pannexin-1 and -3. The authors determined the structure of pannexin-3, revealing two-compartment vestibule organisation inside the channel pore. The authors also solved the structures of two mutants of pannexin-1, R217H (a congenital mutant) and a double mutant W74R/R75D (designed to mimic the pore of pannexin-2). Both mutations appear to constrict the pore, and the authors' experiments suggest that ATP interaction is affected in the mutated proteins.

The study is definitely of interest to the ion channel field. The structure of pannexin-3 is new, and the mutated pannexin-1 versions are quite interesting. Reading through this manuscript I could not get rid of the impression that I am actually reading two separate manuscripts (possibly even three) - one about pannexin-3 structure, one about the congenital mutant, and one about the PANX1dm mutant. The authors may yet find a solution to combine these three parts of the manuscript into a congruent story. In places my impression is that the interpretations are not quite accurate and/or can be improved. The level of detail in the descriptions of the procedures can be improved. The controls should be included to judge the quality of the electrophysiology data - which will then help with the interpretations.

In my opinion, there are three major points that the authors should find a way to address:

1. A proof that PANX1dm is indeed a model of PANX2 - the authors say it is a mimic of PANX2, but there is no functional experiment with PANX2.

2. The ability of CBX to still inhibit the current of PANX1dm is highly intriguing - the authors basically just leave it there. This seems to be the most intriguing finding of the whole study, and an opportunity to explore this some more. Minimally, the authors could attempt to measure and compare the CBX affinities using MST, as they have done for ATPgS.

3. Control experiments should be added: experiments with PANX2, experiments with the mock-transfected or untransfected cells (electrophysiology). Without this it is hard to judge what we are looking at in the ephys experiments - with CBX in particular (related to point 2 above).

An added “bonus” point is the relatively low resolution of each reconstruction - each structure was determined using a rather small number of micrographs. The authors can likely improve the resolution simply by collecting more data - and this will make their manuscript stronger.

Below is a list of issues that would need to be addressed:

The title and the discussion suggest something about “dynamics” - but the authors do not really perform any experiments that address protein dynamics. They also do not perform any simulations of molecular dynamics to warrant the use of the word “dynamics” in relation to this manuscript, neither in the title nor elsewhere in the description of the results/interpretations.

Given a number of available PANX1 structures, the manuscript could benefit from some comparisons - presumably the conditions used by the authors were somewhat different from what was used in the other studies. A supplementary figure describing the differences / similarities would be useful (comparing models and maps).

Page 4 - “..PANX1 in sequence identity and ..” - please rephrase.

Page 4 - “PANX.. sequence identity of 27%” - compared to what?

Page 4, first paragraph of results, second sentence - references are missing. Should this not be in the introduction?

Page 4 - “cryo-EM in the presence of ATP” - why actually was this done in the presence of ATP? Where in the methods section do we find any details about this? Was ATP used, or rather ATPgS? Was the pH of the ATP stock adjusted? At what concentration and at what stage was it added to the sample?

Page 4 - "with a similar transmembrane height" - what does this mean? Does this refer to the length of the TM helix, or to the thickness of the lipid bilayer? It would generally help the reader to see the lipid bilayer margins indicated in the figures (so far the authors do it only in the figure 1A, indicating the TM region).

Page 5 - "A comparison with the Alphafold2.." - where is this comparison? A reference to a publication or to a figure is missing.

Page 6 - why did the authors pick POPE as a lipid to place into their model? Is it necessary and reasonable to do this, considering the relatively low resolution of the reconstruction?

Page 6 - ATPgS - why use ATPgS and not ATP proper? If the authors used ATPgS here and throughout, they should correct all of their figures where they indicate "ATP" as the ligand.

I would be very keen to see whether ATP shows a similar affinity to ATPgS - this is a simple experiment and the authors should show that their choice of ATPgS is valid. The non-hydrolysable nucleotide analogues may behave differently with respect to pannexins, compared to ATP.

Page 7 - the observed weaker affinity of PANX3, compared to PANX1 is really neither explained nor explored further. What does this actually mean? Is the lower apparent affinity equal to lower ability to release ATP? Could it be the other way around? The transported solutes may not need to have high affinity to the protein - it may in fact be advantageous for the solutes to have lower affinity for the transport protein, which should ensure the release of the solute.

Page 7 - 3.9Å .. refined to a resolution of 3.75 Å - this is confusing. What is the resolution of the final refined map?

Page 8 - ""which can be correlated well.." - related to point about the affinity - I am not convinced that this is a clear correlation. Perhaps the authors need to explain this better.

Page 8 - ".. that face the vestibule interior on ATP interaction propensities" - complicated sentence, please rephrase.

Page 8 - "Although the substitution is in TM3.." - which substitution?

Page 9 - “.. structures done so far..” - please rephrase.

Page 9 - ZfPANX1 or zfPANX1?

Page 9 - “checked its binding affinity” - what does “its” refer to?

Page 9 - the choice of the mimicking residues W74 and R75 is not clearly explained. These loops contain other residues that are varied between the two PANX homologues - could it be that mutating more residues could produce an even better PANX2 mimic?

Could it be that the PANX1dm mutant is not really a PANX2 mimic functionally? The authors reasoning is largely based on the sequence - but is there any indication that the pore of the dm mutant behaves functionally in a similar way as the pore of PANX2? It would seem appropriate to perform a comparison using electrophysiology experiments, as in Extended Data Figure 5.

Page 10 - “weakened Kd value of 82 uM” - what does it mean? Currently I am not sure I understand what this means, other than that it is a curious fact.

Possibly the most important observation from the PANX1dm mutant - it still reacts to CBX! The effect is less pronounced, compared to the wild-type PANX1, but it is still there. The presumed “primary binding site” of CBX, W75, is mutated to an arginine, and the drug is still able to bind. What does this mean for our current understanding of how CBX inhibits these channels? The authors run a slight risk of overinterpreting the data with the PANX1dm mutant, since they do not know whether it truly functionally or structurally mimics PANX2 (no data currently in the manuscript). However, this channel seems to be functional, and is inhibited by CBX, despite the mutation of a key residue involved in CBX binding - I think this may be the most important finding presented in this manuscript, not yet fully unpacked by the authors.

Related to this, I am surprised the authors do not show the MST data with CBX. If they have not yet performed those experiments, they should probably perform them. These seem to be relatively simple assays.

Page 10 - “..network with TM..” - “within TM”?

Page 10 - “.. through lipid interactions.” - what does this mean?

Page 10 - “wide pore” or “large pore”?

Page 11 - “.. thereby suggesting that PANX2 channel .. can be very distinct..” - this is a speculation not based on experimental evidence. Some experiments on PANX2 would be useful, as noted above, to provide a comparison and to establish whether the dm mutant can indeed serve as a functional model of PANX2.

Page 11 and Fig 5a-e - analysis of the pore profiles. Why is this here and not in results. And what are the implications of this analysis. This is not quite clear.

Page 12, last paragraph: “..organization.. organization.” - seems somewhat repetitive.

Fig. 1b-c - what do these scale bars correspond to?

Figure 1e and Figure 3g supposedly show PANX1 - but the electrostatic potential colours are totally different (while the scale is the same, -5 to +5). Why is that? What are we actually looking at there? This seems to be an error and should be corrected.

Fig. 3d and 4c - a comparison to PANX1 wt would be useful (as in 3e).

Fig. 3f - it should be fair to conclude that the affinity can not be adequately determined here (even if the software can produce some Kd value, clearly it can not be accurate based on the curve that has not reached even a suggestion of saturation).

Fig. 5 - the colouring of PANX3 is quite different from the rest - why is that?

Extended Data Fig. 5 - please perform / provide the data with the negative controls - without that it is impossible to judge whether the observed currents mean anything at all.

Extended Data Fig. 2 and 7 - are the values for number of micrograph correct in each case, or are these approximate values? Please indicate the exact values, and update the table S1.

Supplementary Fig S2 - it would be great not to obscure the gel with labels. Introduce the labels outside of the gel lanes.

Table S1 - can the authors confirm that pixel sizes were identical for the FalconIII and for the K2 camera?

Table S1 - why not indicate the number of micrographs as well?

Table S1 title - "Cryo-EM .."

Reviewer #3 (Remarks to the Author):

Summary:

The authors solved the first (cryo-EM) structure of human pannexin 3 (PANX3) – an ion channel with a large pore to release cellular ATP upon activation. In comparison with PANX1, PANX3 reveals a second compartment in its vestibule. The authors also report the structures of the PANX1 congenital mutant R217H along with the PANX1 pore mutant W74R/R75D which mimics function of PANX2. The congenital mutant and the PANX2-mimic induce structural changes that lead to a partially closed pore and altered ATP-binding. The authors suggest that channel conductance (which they did not measure) of the congenital mutant displays weakened voltage sensitivity.

Review:

The structural work overall appears to be reasonable, although some of the interpretations should be made with much more caution or avoided entirely given the relatively low resolution of all three structures (details given below).

The authors use the term "dynamics" in the title, abstract and throughout the text. This is very misleading because they do not show any dynamics and instead only different pore architectures

dependent on the amino-acid composition of their protein. The authors should either use an appropriate term (“geometry” or something like this) or run MD simulations of their structures and make assessment of the corresponding dynamics.

The authors make a big deal about the “second vestibule in the neck region”. What is this second vestibule? There is no illustrations or labels of this vestibule! By looking at the structural comparison of PANX1 and PANX3 in Fig. 1, these two channels look grossly similar!

Kd measurement for ATP(analog)-binding have high errors and should therefore also be interpreted a lot more carefully, if at all (details given below).

Current density in electrophysiological recordings was used as a correlate with the pore radius. Given that the authors do not explain their procedure of current normalization, they likely simply measure the current amplitude. However, the current amplitude might be larger for WT channels simply because they express better. The authors need to parallel their electrophysiological recordings with measurement of the surface expression. Alternatively, the authors should measure single channels and estimate conductance directly.

Structural work:

“The main interactions at the pore in PANX1WT are W74-R75 cation- π interaction within the subunit and R75-D81 interaction between two subunits. As these residues were mutated, the interactions are lost in the PANX1DM.” At 4.3 Å resolution (looking at the FSC curves, likely less than that), I wonder whether these interactions can be confidently interpreted at all?

A suggestion for resolution improvement: have the authors attempted focussed classification on the ICD (and maybe TMD)? This would be one way to address the drastic worsening of density quality there.

Functional work:

Fig. 2e,f: the error in Kd for ATP-binding in PANX3 is substantial. Given the high error, the Kds for PANX1 and 3 could be not that far apart after all. Is it possible to improve the measurements/reduce the error range? The same applies to Extended Data Fig. 6c,e – especially in c where the error is almost as large as the mean value.

Same is true for the Kd value for the PANX1 DM which has an error almost as large as the determined mean Kd. It is challenging to compare Kds in this case.

Kd value data should either be improved (more/better measurements) or the interpretations in the manuscript should be adjusted.

Minor issues:

Page 3, first sentence of the 3rd paragraph. Two papers that report the first structures of Pannexin 1 are cited (refs. 16 and 18). How about other papers, for example: <https://elifesciences.org/articles/54670> and <https://www.nature.com/articles/s41422-020-0298-5>?

Page 3, second sentence of the 3rd paragraph. What large-pore ion channels are mentioned here? Examples? References?

The first paragraph of Results (see also page 7, second paragraph). "Clear density ... allowed accurate modeling of the side chains..." Illustration in EDFig. 3a does not allow to see any side chains. The authors should either make an illustration with clear side chain densities or correct the text, as it is highly unlikely to "accurately" model side chains at 3.9 Å resolution.

EDFig. 3b shows density for a lipid. What kind of structure does this density belong to? Why did the authors use POPC to model this density? Did they add POPC during purification and forgot to mention this?

Page 5, first paragraph. "The negatively charged surface may facilitate the role as a calcium channel". What about the positively charged surface in the same pore? May it prevent PANX3 to play the role of calcium channel? The authors need to run MD simulations to decide on which of the surfaces is more important for permeation.

Page 5, second paragraph. The authors claim that the "C-terminus is unlikely to serve as a plug to block access to the channel" and "unlikely to play a role in channel opening". The authors have to create the C-terminally truncated mutant (d373-392) and prove their statements functionally.

Page 6, 3rd paragraph. “The TM1-TM2 linker adopts alpha-helical conformation in PANX3, instead of a loop in PANX1”. This is not at all clear from Fig. 2d. Besides, this statement contradicts EDFig. 4a!

Fig. 2g. What is the shown current normalized to?

Page 7, second paragraph. What does it mean: voltage-sensitive currents at positive potentials? Currents are always sensitive to voltage according to the Ohm’s law!

Page 8, second paragraph. “W74 shifts by ... an angle”. What does this mean?

Figure 3f. The concentration dependence does not show saturation at high concentrations. The concentration range should be extended to high concentrations for fitting to be done correctly.

As far as the manuscript and data presentation is concerned, at times, the chosen illustrations do not help the reader at all to interpret the data, nor do they support the written text in the manuscript. I usually do not comment on the style of illustrations, however, in this case some of the figures simply do not support the written text, an assessment of which becomes therefore unfeasible without access to density maps and corresponding models. I am providing some details below:

This sentence “Two disulfide bonds stabilize the extracellular domains, C66- C261(SS1) and C84- C242(SS2), that form between EL1 and EL2 (Extended Data Fig. 4a, b)” uses too many undefined abbreviations and it is not clear what the authors are trying to say or show here. I would suggest to add a panel to Extended Data Fig. 4 to visualize these interactions and also to rephrase this sentence with clear explanations (what are EL1 and EL2, for instance?).

“N-linked glycosylation at the N255 position in the EL2 of PANX1 was implicated in preventing the formation of gap junctions.” Reference(s) needed here.

The authors claim that they observe density for N-acetylglucosamine (NAG) at the predicted glycosylation site in their PANX3 structure. The corresponding panel in Fig 1f. is not self-evidently showing this. Without access to the maps, it is hard to make a judgement here. I would suggest to show an additional close-up of density with the modelled molecule inside it (like POPE in Extended Data Fig. 3 for instance).

“A comparison with the Alphafold2 model of PANX3 with the experimental model in this study indicates that the N-terminus faces the cytosol.” I think the authors assume a good pannexin structural knowledge from the future readers here. The mentioned Alphafold2 models needs to be shown in the manuscript in comparison with the presented structural models. Please either add a figure in Extended Data or clearly refer to an existing figure such as Fig 1d.

Regarding the first two paragraphs/sections of “PANX3 displays a double-sieve pore organization”: the residue depiction in Fig. 2a is unfortunate, especially if cation- π interactions are discussed here, please provide a better, more detailed illustration (no need to show the whole channel here, Fig. 1 showed this already but side chains and densities for them should). Same applies to the salt bridge and hydrogen bond discussed next; please provide an illustration. The cation- π interaction is not “altered” by W74I, it is “eliminated” because I74 does not have pi-orbitals!

The same problem arises with POPE: where is this lipid situated in the map/model? It is completely out of context if no structural/spatial reference is given.

Fig. 2a,b: it took me some time to realise that these panels relate to each other.

The interactions that the authors are trying to discuss are not facilitated at all by the way Fig. 2d is currently designed/depicted. Please consider to show more details/residue interactions here in correspondence with the text.

Again here: “The R75 residue interacts with the D81 of the adjacent subunit through a salt bridge, consistent with PANX1WT.” -> where am I supposed to look? No figure provided.

I would mention the 42% sequence identity between PANX1 and 3 in the introduction already, if possible, to provide context for the 27% sequence identity between PANX1 and 2 which is discussed after that.

Response to Reviewers' comments

We thank all the three reviewers for their suggestions and critical reading of the manuscript.

Reviewer #1 (Remarks to the Author):

The manuscript by Hussain et al. describes the structural comparison between Pannxin1, PANX1 mutant, PANX2 mimetic Panx1 mutant, and Panx3. PANX is the major large-pore channel permeable to small anions (Cl⁻) and ATP. The Panx family comprises three proteins, Panx1-3. Panx1 is the most dominantly expressed protein studied extensively, whereas Panx2 and Panx3 are understudied. Several Panx1 structures were published in 2020, starting with Michalski et al. (eLife 2020). The Panx1 structure showed narrow constrictions at the extracellular region and another semi-narrow constriction in the TMD. In this study, the authors made Panx1 mutant W74R/R75D that mimics Panx2, and the Panx1 R217H mutant. Other groups posted the Panx2 and Panx3 structures in BioRxive but have not published them yet. The major weakness of this manuscript is that the authors do not clearly explain their structures and structure-based mechanisms. For example, why does Panx3 have a lower affinity to ATP?

Response: In the revised manuscript, we have addressed the reviewer's concerns by incorporating additional descriptions about the ATP binding and references that were missed in the previous version of the manuscript. We would like to inform the reviewer that the BioRxiv preprint on PANX3 belongs to us, and we have so far not observed any other group reporting a PANX3 and/or a PANX2 structure either as a preprint or as a published article.

We have, at various locations in the manuscript (described below), enhanced the discussion of the results in the manuscript as suggested by the reviewer. Also, given the absence of binding site information on ATP interactions in the PANX1 or 3 structures; despite multiple structures with a better resolution, it would be a difficult proposition to conclude as to why PANX3 could have a lower affinity to ATP. Any discussion in this context would be mere speculation.

Why do they have lower sensitivity to CBX? They state that the substitution of W74 causes that, but how? The messages are not clear. Overall, this study provides structures but not mechanisms. Furthermore, several citations need to be added appropriately to improve the level of accuracy of this manuscript.

Response: We have discussed the lowered sensitivity to CBX in the discussion. While the current understanding of carbenoxolone implicates the role of pore lining residues in dictating binding, its sterol architecture could allow it to interact at alternate sites on the channel that are occupied by lipid molecules. This has been added to the PANX3 discussion in "The ECD region, particularly loop1, surrounding the pore was previously implicated as being important for CBX interactions. The substitutions among pore lining residues observed in PANX3 could be a reason for the reduced sensitivity to CBX (Extended Data Fig. 6a). The ability of CBX to interact with PANX3, albeit weakly, despite a wider pore, reinforces the idea that its interactions at extracellular domain can modulate channel activity instead of directly acting

as an asymmetric pore blocker. Alternate uncharacterized sites for CBX interactions can exist within PANX isoforms given its sterol like chemical structure that modulates channel activity. We also tried to get carbenoxolone interaction affinities through MST binding studies on PANX1 but we did not observe saturation suggesting non-specific interactions in addition. The specific location of carbenoxolone binding requires detailed investigation at higher resolution in comparison to the ones obtained in this study. As per the reviewer's suggestion, we have more extensively cited the literature that was missing in the earlier draft of the manuscript.

Specific points:

-The title is misleading since this work does not address any dynamics. The work is strictly on the structural comparison.

Response: We appreciate the concern of the reviewer in this context.

We have altered the title: "Structural insights into the organization and channel properties of Pannexin isoforms 1 and 3". We have reorganized the focus of the manuscript on pannexin1 and 3 and less so on pannexin 2 mimic in comparison to the previous version.

-While providing the field with Panx2 and 3 structures is important, the authors must state why they worked on it. For example, are there any specific functional features of Panx2 or 3 the authors want to explain? If so, why are they important in the context of general biology? It should be stated in the manuscript.

Response: Although PANX1 is a more widely studied protein compared to the other two isoforms, these isoforms have different tissue localization and do not compensate for the loss of one isoform. The following statements are added in the introduction to strengthen the arguments for this study, "PANX3 is expressed in osteoblasts, chondrocytes, and skin and plays a major role in calcium homeostasis suggesting an independent functional niche. Despite the high sequence conservation with its isoform, PANX3 displays distinct structural features, localization, and functional characteristics compared to PANX1_{WT}." "The PANX isoforms, PANX1, PANX2, and PANX3 harbour substitutions in the residues that control channel gating and could have architectural differences that form the basis for altered properties among the three isoforms." This is the first report where PANX3 currents are being reported and could be a study that will nucleate future investigations into the role of pannexin isoforms 1 and 3. Pannexin 2 mimic is not stressed on in the manuscript and therefore we have not described its biological roles.

-What are the functional states of the cryo-EM structures? The authors need to explain. Given that the C-terminus is not cleaved by a caspase, could it represent the inactive state?

Response: The functional states of the four structures are as follows

PANX1_{WT} = Open to ions but closed to ATP permeability (similar to other published structures); PANX1_{R217H} = Closed; PANX1_{DM} = Closed; PANX3 = Open state. We do not ascribe this to be an inactive state since multiple

studies show the presence of ATP release upon activation by high extracellular K⁺ ions. (Iwamoto T, et al. (2017), PLOS ONE)

-pg3: Citation 16/19. Detergents are mostly bound to what was published as the CBX binding site. For example, the apo-structure of PANX has the same density around W74.

Response: While we agree that the density at the pore could be an artifact of symmetry averaging to a certain extent, other physiological studies like Michalski *et al.* 2016, Zengqin Deng *et al.* have demonstrated that mutation at W74 in loop1 compromises carbenoxolone binding as observed in physiological experiments. We have modified our arguments accordingly.

-pg 6: With respect to the observation that positively charged patches are present, stating that it may be related to ATP binding and release is casual. It has to be experimentally proven.

Response: We would like to draw the reviewer's attention to the fact that we have indeed performed ATP binding experiments of mutations around cationic patches (residues K24 and R128, Extended data figure 8; Suppl Fig. S3). The R128 particularly displays a major compromise of ATP binding, suggesting the importance of cationic residues in specific regions of the channels for ATP binding. Although we tried to correlate the binding with ATP release the experiments proved to be highly inconsistent even for PANX1_{wt} even when tried in multiple cell lines.

-pg6: ATP binding assay with MST – 1) it would be necessary to show traces from the actual experiments, especially since the signals in MST experiments vary. 2) The difference of ATP binding on Panx1 and 3. Could the authors address why there are differences in their structures??

Response: We have included raw data points to perform the binding analyses. We have shown this data for three independent replicates and include all the fits in supplementary figure S3. The panels display raw data and no additional normalization was performed. We have also performed ATP binding assays with PANX1 to show that the affinity between ATP and ATP- γ S are similar at 21 μ M and 13 μ M. For better clarity, traces for three independent experiments have been shown as part of the supplementary figure 3. The differences between ATP binding affinities among PANX1 and PANX3 cannot be answered till a clear binding site is established.

-Some citations need to be added or corrected.
1) Pg 3: The Syrjanen and Michalski review (2021 JMB) wraps up the recent progress in the large-pore channel field and should be a useful one to cite.
2) pg5: CALHM citation. Syrjanen *et al.* (NSMB, 2020) is the first paper to show the diversity in the oligomeric state and the mechanism to control it.
3) Michalski and Syrjanen *et al.* (eLife 2020) must be cited in the introduction since that represents the first structural paper on PANX.

Response: We regret the omission of the above three references and have now added them to the manuscript at appropriate locations.

Reviewer #2 (Remarks to the Author):

The manuscript by Hussain et al, "Structural insights into pore dynamics of human PANX isoforms" describes a structural and biochemical/biophysical study of two PANX ion channels, PANX-1 and -3. The authors determined the structure of PANX-3, revealing two-compartment vestibule organisation inside the channel pore. The authors also solved the structures of two mutants of PANX-1, R217H (a congenital mutant) and a double mutant W74R/R75D (designed to mimic the pore of PANX-2). Both mutations appear to constrict the pore, and the authors' experiments suggest that ATP interaction is affected in the mutated proteins.

The study is definitely of interest to the ion channel field. The structure of PANX-3 is new, and the mutated PANX-1 versions are quite interesting. Reading through this manuscript I could not get rid of the impression that I am actually reading two separate manuscripts (possibly even three) - one about PANX-3 structure, one about the congenital mutant, and one about the PANX1_{dm} mutant. The authors may yet find a solution to combine these three parts of the manuscript into a congruent story. In places my impression is that the interpretations are not quite accurate and/or can be improved. The level of detail in the descriptions of the procedures can be improved. The controls should be included to judge the quality of the electrophysiology data - which will then help with the interpretations.

Response. We thank the reviewer for considering this to be an interesting study. We have tried our best to make the manuscript coherent and incorporated most of the reviewer's comments, and/or tried addressing the concerns through text changes. We have suitably altered the title and minimized the focus on PANX2 to make this coherent manuscript and highlight the PANX1_{DM} structure to highlight the effects of charged substitutions on PANX1 pore.

In my opinion, there are three major points that the authors should find a way to address:

1. A proof that PANX1_{dm} is indeed a model of PANX2 - the authors say it is a mimic of PANX2, but there is no functional experiment with PANX2.

Response: We moderate our arguments in the manuscript that the PANX1_{DM} is a mimic of the PANX2 channel. Heterologous expression of PANX2 in HEK cells yields a poor-quality protein that obviates the possibility of performing good-quality recordings. For reference, we have added a FSEC comparison of PANX1, 2, and 3 (below).

Since we could not perform a PANX2 recording, we seek to modify our claims about the PANX1_{DM} as a pore mimic of PANX2 and suggest it as a mutation that constricts the pore of the PANX1 channel.

2. The ability of CBX to still inhibit the current of PANX1_{dm} is highly intriguing - the authors basically just leave it there. This seems to be the most intriguing finding of the whole study, and an opportunity to explore this some more. Minimally, the authors could attempt to measure and compare the CBX affinities using MST, as they have done for ATPgS.

Response: We thank the reviewer for the suggestion. We have included an additional statement at this juncture as follows, “It is rather interesting that CBX retains minimal interactions with the PANX1_{DM} despite major substitutions in the pore. This further indicates that presence of alternate sites of interaction that could modulate channel gating.” We further attempted to measure the binding affinity of CBX with PANX1_{WT} using MST. However, we did not get a proper saturation even at a higher concentration of CBX (5 mM), making it unfeasible to compare the binding affinity of PANX1 mutants. For reference, we have included the MST analysis of CBX in Supplementary figure 2. (pls see below)

3. Control experiments should be added: experiments with PANX2, experiments with the mock-transfected or untransfected cells (electrophysiology). Without this it is hard to judge what we are looking at in the ephys experiments - with CBX in particular (related to point 2 above).

Response: We thank the reviewer for the suggestion. Due to difficulties with PANX2 expression, we will not be adding this experiment. Control experiments with untransfected cells have been added in Extended data figure 7 and main figures 2i and 3d.

An added "bonus" point is the relatively low resolution of each reconstruction - each structure was determined using a rather small number of micrographs. The authors can likely improve the resolution simply by collecting more data - and this will make their manuscript stronger.

Response: Although we cannot commit to getting better resolution structures at this point due challenging sample preparation and limited microscope time, we do intend to try this as part of the revision of the manuscript particularly for PANX3.

Below is a list of issues that would need to be addressed:

The title and the discussion suggest something about "dynamics" - but the authors do not really perform any experiments that address protein dynamics. They also do not perform any simulations of molecular dynamics to warrant the use of the word "dynamics" in relation to this manuscript, neither in the title nor elsewhere in the description of the results/interpretations.

Response: We acknowledge this issue and have altered the title suitably.

The revised title as stated earlier is, "Structural insights into the organization and channel properties of Pannexin isoforms 1 and 3."

Given a number of available PANX1 structures, the manuscript could benefit from some comparisons - presumably the conditions used by the authors were somewhat different from what was used in the other studies. A supplementary figure describing the differences / similarities would be useful (comparing models and maps).

Response: We have enhanced the number of comparisons by comparing our structure with the hPANX1 structure (6wbf) which shows a near identical overlap (Suppl. Fig S1), and also with the ZfPANX1 structure (6v6d) determined by Michalski *et al.* We also draw a structural comparison of PANX1 with PANX3; and PANX1 with the AlphaFold2 model of PANX2 in Extended data figure 5d and 5e.

Page 4 - "..PANX1 in sequence identity and .." - please rephrase.

Response: We have rephrased the sentence. "PANX3 has a sequence identity of 42% with PANX1 and is the shortest among the three isoforms with a length of 392 residues."

Page 4 - "PANX.. sequence identity of 27%" - compared to what?

Response. The new sentence reads as follows, " PANX2 is the most divergent among the three, with a sequence identity of 27% with PANX1,.."

Page 4, first paragraph of results, second sentence - references are missing. Should this not be in the introduction?

Response: The sentence has been moved to the introduction, and appropriate references have been added.

Page 4 - "cryo-EM in the presence of ATP" - why actually was this done in the presence of ATP? Where in the methods section do we find any details about this? Was ATP used, or rather ATPgS? Was the pH of the ATP stock adjusted? At what concentration and at what stage was it added to the sample?

Response: We performed this in the presence of ATP in the hope that we could obtain information on an ATP binding site, as PANX3 has also been shown to release ATP in the presence of high extracellular potassium. Iwamoto T *et al.* 2017, PLOS ONE

ATP at a final concentration of 1 mM was added to the solution 30 minutes before freezing the grids. The stock of ATP was made in the SEC buffer (Tris pH 8.0). We have included these details in the method section.

Page 4 - "with a similar transmembrane height" - what does this mean? Does this refer to the length of the TM helix, or to the thickness of the lipid bilayer? It would generally help the reader to see the lipid bilayer margins indicated in the figures (so far the authors do it only in the figure 1A, indicating the TM region).

Response: We have included the detergent micelle in Figure 1a and b. This provides a clear demarcation of the membrane bilayer that is indicated in the figure panels.

Page 5 - "A comparison with the Alphafold2.." - where is this comparison? A reference to a publication or to a figure is missing.

Response: The Alphafold2 structural comparisons are included in Extended data figure 5d and e.

Page 6 - why did the authors pick POPE as a lipid to place into their model? Is it necessary and reasonable to do this, considering the relatively low resolution of the

reconstruction?

Response. The presence of phospholipid during detergent extraction is observed in the higher resolution structure of PANX1. The observed lipid in the map likely corresponds to a phospholipid that could be extracted during detergent treatment with GDN. We have modeled POPE in the density. The density for POPE is shown in the Extended data figure 5c.

Page 6 - ATP γ S - why use ATP γ S and not ATP proper? If the authors used ATP γ S here and throughout, they should correct all of their figures where they indicate "ATP" as the ligand.

Response. We thank the reviewer for the input. We have modified the figures. We initially compared the binding affinity of PANX1 with both ATP and ATP- γ S. The binding affinities are within the error range ($21 \pm 11 \mu\text{M}$ and $13 \pm 3 \mu\text{M}$ for ATP and ATP- γ S, respectively). The ATP- γ S is a non-hydrolyzable analog, and MST measurements are more consistent as compared to ATP, which has a higher hydrolysis tendency.

As these experiments can take time, we used ATP- γ S to avoid differences in the pH and ATP hydrolysis during the experiment.

I would be very keen to see whether ATP shows a similar affinity to ATP γ S - this is a simple experiment and the authors should show that their choice of ATP γ S is valid. The non-hydrolysable nucleotide analogues may behave differently with respect to PANXs, compared to ATP.

Response: The comparison between ATP and ATP- γ S binding was done for PANX1 wild-type, and the affinities are $21 \mu\text{M}$ and $13 \mu\text{M}$, respectively. We have included the graphs as a supplementary figure 1. The comparison is also shown in the previous comment.

Page 7 - the observed weaker affinity of PANX3, compared to PANX1 is really neither explained nor explored further. What does this actually mean? Is the lower apparent affinity equal to lower ability to release ATP? Could it be the other way around? The transported solutes may not need to have high affinity to the protein - it

may in fact be advantageous for the solutes to have lower affinity for the transport protein, which should ensure the release of the solute.

Response: As pointed out for reviewer 1, since we do not have ATP release experiments to correlate with ATP binding we may not be able to highlight the significance of ATP binding differences between PANX3 and PANX1. The lower ATP- γ S binding affinity of PANX3 could be because of the altered electrostatics of the pore compared to PANX1 (Figure 1d). As we have seen in PANX1 mutants, charge alteration leads to lower ATP binding (K24A, R128A, R75A, Extended data figure 8c-f). As a proper binding site is yet to be determined, it remains difficult to speculate the reason behind this observation.

We have included this in the concluding statement of section 2 dealing with PANX3.

Page 7 - 3.9 Å .. refined to a resolution of 3.75 Å - this is confusing. What is the resolution of the final refined map?

Response: The resolution of PANX3 is 3.9 Å, and the resolution of PANX1_{WT} is 3.75Å.

Page 8 - "" which can be correlated well.." - related to point about the affinity - I am not convinced that this is a clear correlation. Perhaps the authors need to explain this better.

Response: We thank the reviewer for the suggestion and have removed this from the manuscript.

Page 8 ".. that face the vestibule interior on ATP interaction propensities" - complicated sentence, please rephrase.

Response: Thank you for the suggestion. We have rephrased the sentence. The modified sentence is as follows: "that face the vestibule of the channel and can influence ATP interactions."

Page 8 - "Although the substitution is in TM3.." - which substitution?

Response: We have rephrased the sentence.
"Although the R217H substitution is in the TM3"

Page 9 – ".. structures done so far.." – please rephrase.

Response: We have rephrased the sentence.

Page 9 - ZfPANX1 or zfPANX1?

Response. Changed to ZfPANX1

Page 9 - "checked its binding affinity" - what does "its" refer to?

Response: It refers to PANX1_{R217H}. We have now rephrased the sentence for clarity.

Page 9 - the choice of the mimicking residues W74 and R75 is not clearly explained. These loops contain other residues that are varied between the two PANX homologues - could it be that mutating more residues could produce an even better PANX2 mimic?

Response: We have altered and provided an improved argument for the PANX1_{DM} as follows, "Altering residues around the PANX1 constriction point can alter the properties of the channel as observed in multiple studies. Single substitutions of W74 and R75 in the pore region cause an incorrect assembly of PANX1. In order to study the effects of a double charged substitution in PANX1_{WT}, we created a PANX1 double mutant (PANX1_{DM}) by substituting W74 and R75 with arginine and aspartate, respectively. These substitutions are similar to the pore residues of PANX2 revealed by multiple sequence alignment. Surprisingly, we obtained a minor fraction of well assembled PANX1_{DM} channel whose structure of PANX1_{DM} was elucidated to a resolution of 4.3 Å (Table S1)."

Mutating more residues is unlikely to work as the difference in residues between the PANX1 and PANX1_{DM} is rather extensive. We have substantially altered the manuscript to reduce the issue of PANX1_{DM} being a mimic of PANX2.

Could it be that the PANX1_{dm} mutant is not really a PANX2 mimic functionally? The authors reasoning is largely based on the sequence - but is there any indication that the pore of the dm mutant behaves functionally in a similar way as the pore of PANX2? It would seem appropriate to perform a comparison using electrophysiology experiments, as in Extended Data Figure 5.

Response: It is quite likely that PANX1_{DM} does not mimic PANX2 functionally. It is also possible that a higher oligomeric arrangement predicted for PANX2 might overcome the narrow constriction formed in the heptameric PANX1_{DM}.

Page 10 - "weakened K_d value of 82 μM" - what does it mean? Currently I am not sure I understand what this means, other than that it is a curious fact.

Response: The PANX1_{DM} mutant shows a lower affinity for ATP compared to PANX1_{WT}. We have rephrased the sentence for better clarity which reads as follows, "The ATP binding data shows a weakened K_d value of 78 μM compared to ATP binding affinity for PANX1_{WT} (13 μM)".

Possibly the most important observation from the PANX1_{dm} mutant - it still reacts to

CBX! The effect is less pronounced, compared to the wild-type PANX1, but it is still there. The presumed "primary binding site" of CBX, W75, is mutated to an arginine, and the drug is still able to bind. What does this mean for our current understanding of how CBX inhibits these channels? The authors run a slight risk of overinterpreting the data with the PANX1_{dm} mutant, since they do not know whether it truly functionally or structurally mimics PANX2 (no data currently in the manuscript). However, this channel seems to be functional, and is inhibited by CBX, despite the mutation of a key residue involved in CBX binding - I think this may be the most important finding presented in this manuscript, not yet fully unpacked by the authors.

Related to this, I am surprised the authors do not show the MST data with CBX. If they have not yet performed those experiments, they should probably perform them. These seem to be relatively simple assays.

Response: We have added the following statement to highlight this point made by the reviewer in page 10, "It is rather interesting that CBX retains minimal interactions with the PANX1_{DM} despite major substitutions in the pore. This further indicates that presence of alternate sites of interaction that could modulate channel gating." Accurate binding affinity could not be determined for PANX1_{WT} as we were unable to get a proper saturation even at higher concentrations of CBX. We have included the MST analysis with CBX with PANX1 in supplementary figure 3b.

Page 10 - "..network with TM.." - "within TM"?

Response: We have modified the sentence.

Page 10 - ".. through lipid interactions." - what does this mean?

Response: The reference to lipid interactions is removed from the text.

Page 10 - "wide pore" or "large pore"?

Response: Large-pore. The sentence has been modified.

Page 11 - ".. thereby suggesting that PANX2 channel .. can be very distinct.." - this is a speculation not based on experimental evidence. Some experiments on PANX2 would be useful, as noted above, to provide a comparison and to establish whether the dm mutant can indeed serve as a functional model of PANX2.

Response: We have incorporated significant changes in the manuscript and do not delve into the similarity of PANX2 anymore.

Page 11 and Fig 5a-e - analysis of the pore profiles. Why is this here and not in results. And what are the implications of this analysis. This is not quite clear.

Response: We thank the reviewer for the input. We wanted to show a summary side-by-side comparison of the change in the pore radius as an effect of the substituions in PANX3 and PANX1 mutants. As a consequence we made this comparison as the final figure in the manuscript.

Page 12, last paragraph: "..organization.. organization." - seems somewhat repetitive.

Response: We have modified the sentence.

Fig. 1b-c - what do these scale bars correspond to?

Response: These are not scale bars. They represent the width of the PANX1 and 3 at the base and at the extracellular face. We have rephrased the legend for better clarity.

Figure 1e and Figure 3g supposedly show PANX1 - but the electrostatic potential colours are totally different (while the scale is the same, -5 to +5). Why is that? What are we actually looking at there? This seems to be an error and should be corrected.

Response: The comparison was done for a trimmed version of the PANX1_{WT} structure to mimic the residues in PANX1_{R217H}. As the R217H lacked few residues(11-24) in N-terminus, we deleted the corresponding residues in PANX1_{WT} for a better comparison of the surface electrostatics.

Fig. 3d and 4c - a comparison to PANX1 wt would be useful (as in 3e).

Response: We have made the changes and kept the IV curve for PANX1 and its mutants as a separate figure, along with the raw data, for a better comparison. (Extended Data figure 7)

Fig. 3f - it should be fair to conclude that the affinity can not be adequately determined here (even if the software can produce some K_d value, clearly it can not be accurate based on the curve that has not reached even a suggestion of saturation).

Response: We have optimized the ATP- γ S concentration for the MST experiments with respect to PANX1_{WT}. The highest ATP- γ S concentration of 2mM showed a proper saturation and a consistent binding affinity within three independent replicates for PANX1_{WT}. Similar conditions were used for the mutants to rule out different variables that might play a role at higher concentrations of ATP- γ S.

For example, at higher concentrations of ATP- γ S, we observed a significant change in the pH, most likely due to hydrolysis.

However, if required, we can attempt to repeat the experiments with higher concentrations of ATP to get a proper saturation during the revision of the manuscript.

As pointed out by the reviewer, a proper saturation did not reach to determine the binding affinity for the PANX1_{R217H} mutant accurately, therefore, we had implied it as an apparent affinity in the manuscript which shows the altered behavior of the mutant as compared to the PANX1_{WT}.

Fig. 5 - the colouring of PANX3 is quite different from the rest - why is that?

Response: Since PANX3 is a novel structure we colored the protomers in different colors.

Extended Data Fig. 5 - please perform/provide the data with the negative controls - without that it is impossible to judge whether the observed currents mean anything at all.

Response: The changes are incorporated in the figure. Data with untransfected controls is included. (Extended Data Fig. 7)

Extended Data Fig. 2 and 7 - are the values for number of micrograph correct in each case, or are these approximate values? Please indicate the exact values, and update the table S1.

Response: Exact values have been indicated.

Supplementary Fig S2 - it would be great not to obscure the gel with labels. Introduce the labels outside of the gel lanes.

Response: The figure is appropriately modified.

Table S1 - can the authors confirm that pixel sizes were identical for the FalconIII and for the K2 camera?

Response: The magnification for Falcon III and K2 detectors was chosen to be the same. In the instrument, Falcon III is before the GIF, and K2 is after the energy filter. Thus, the nominal magnification of 75,000 x used for Falcon III imaging and the 130,000 x used for K2 in EFTEM (Energy filtered TEM) mode are the same, and thus the pixel size.

Table S1 - why not indicate the number of micrographs as well?

Response: We have indicated the micrograph number in Table S1.

Table S1 title - "Cryo-EM .."

Response: We have made the changes.

Reviewer #3 (Remarks to the Author):

Summary:

The authors solved the first (cryo-EM) structure of human PANX 3 (PANX3) – an ion channel with a large pore to release cellular ATP upon activation. In comparison with PANX1, PANX3 reveals a second compartment in its vestibule. The authors also report the structures of the PANX1 congenital mutant R217H along with the PANX1 pore mutant W74R/R75D which mimics function of PANX2. The congenital mutant and the PANX2-mimic induce structural changes that lead to a partially closed pore and altered ATP-binding. The authors suggest that channel conductance (which they did not measure) of the congenital mutant displays weakened voltage sensitivity.

Review:

The structural work overall appears to be reasonable, although some of the interpretations should be made with much more caution or avoided entirely given the relatively low resolution of all three structures (details given below).

Response: We thank the reviewer for finding our work to be reasonable. We will attempt to be conservative in our interpretations of the results as suggested and accordingly modified the manuscript.

The authors use the term "dynamics" in the title, abstract and throughout the text. This is very misleading because they do not show any dynamics and instead only different pore architectures dependent on the amino-acid composition of their protein. The authors should either use an appropriate term ("geometry" or something like this) or run MD simulations of their structures and make assessment of the corresponding dynamics.

Response: As per the suggestion fo reviewers1 and 3 ,we have now altered the manuscript and do not refer to the word dynamics in the title. The new title of the manuscript is as follows, "Structural insights into the organization and channel properties of PANX isoforms 1 and 3." This title, we hope, would be more acceptable to the reviewers. Since we have not been able to do MD simulations in this study we decided to remove any reference to the word dynamics in this study.

The authors make a big deal about the "second vestibule in the neck region". What is this second vestibule? There is no illustrations or labels of this vestibule! By looking at the structural comparison of PANX1 and PANX3 in Fig. 1, these two channels look grossly similar!

Response: In figure 1, we highlight the region of the vestibule in the upper half of PANX3 that looks distinct from PANX1 both in electrostatic surface and also in structural features. The helical region around F58 in PANX3 induces this constriction, whereas PANX1 has an unwound helix in the region (Fig. 2d). We have modified the text to remove references to double sieve organization.

K_d measurement for ATP(analog)-binding have high errors and should therefore also be interpreted a lot more carefully, if at all (details given below).

Response: The errors occurred due to the mixing of independently measured data. We have now provided individual repeats of the graphs, and the errors for single data point containing measurements is much lower.

Current density in electrophysiological recordings was used as a correlate with the pore radius. Given that the authors do not explain their procedure of current normalization, they likely simply measure the current amplitude. However, the current amplitude might be larger for WT channels simply because they express better. The authors need to parallel their electrophysiological recordings with measurement of the surface expression. Alternatively, the authors should measure single channels and estimate conductance directly.

Response: Expression was observed to be similar for most of the constructs used in the study using FSEC. The current density (pA/pF) accounts for the surface expression of the channels in HEK293 cells. Similarly conductance density was also used (nS/pF) to assess changes in conductance properties among the PANX1 mutants and PANX3.

Structural work:

"The main interactions at the pore in PANX1WT are W74-R75 cation- π interaction within the subunit and R75-D81 interaction between two subunits. As these residues were mutated, the interactions are lost in the PANX1DM." At 4.3 Å resolution (looking at the FSC curves, likely less than that), I wonder whether these interactions can be confidently interpreted at all?

Response: While we understand the concern of the reviewer, we have surprisingly clear densities at the pore region for the W74R/R75D double mutant. We have included densities for the residues into which side chains were modeled in the study as figures in the manuscript (Figure 4a,b). The density of side chains in the region of interest is indeed clear without serious ambiguities. We are also quite happy to share the maps and coordinates of the deposited structures in a folder shared with the editor. We have included the

densities in the figure below as well.

A suggestion for resolution improvement: have the authors attempted focussed classification on the ICD (and maybe TMD)? This would be one way to address the drastic worsening of density quality there.

Response. The structures reported here are after performing local refinement in cryosparc. We can attempt to improve the maps through masking and focused refinement of TMDs if given a chance to revise the manuscript.

Functional work:

Fig. 2e,f: the error in Kd for ATP-binding in PANX3 is substantial. Given the high error, the Kds for PANX1 and 3 could be not that far apart after all. Is it possible to improve the measurements/reduce the error range? The same applies to Extended Data Fig. 6c,e – especially in c where the error is almost as large as the mean value.

Same is true for the Kd value for the PANX1 DM which has an error almost as large as the determined mean Kd. It is challenging to compare Kds in this case.

Kd value data should either be improved (more/better measurements) or the interpretations in the manuscript should be adjusted.

Response: In all the above cases, the error value increased due to merging separate datasets. We now report a single-point binding curve with lower error values in Kd measurement.

Minor issues:

Page 3, first sentence of the 3rd paragraph. Two papers that report the first structures of PANX 1 are cited (refs. 16 and 18). How about other papers, for example: <https://elifesciences.org/articles/54670> and <https://www.nature.com/articles/s41422-020-0298-5>?

Response. The two references have been included in the manuscript.

Page 3, second sentence of the 3rd paragraph. What large-pore ion channels are mentioned here? Examples? References?

Response: The examples and references have been added. The modified sentence read as:

“The topology is similar to large-pore ion channels such as calcium homeostasis modulators (CALHMs), leucine rich repeat containing VRAC subunit 8 (LRRC8), connexins and innexins with four transmembrane helices

(TMs), a prominent extracellular domain (ECD), and a disordered C-terminus that is not visible in most PANX1 structures¹⁹

The first paragraph of Results (see also page 7, second paragraph). "Clear density ... allowed accurate modeling of the side chains..." Illustration in EDFig. 3a does not allow to see any side chains. The authors should either make an illustration with clear side chain densities or correct the text, as it is highly unlikely to "accurately" model side chains at 3.9 Å resolution.

Response: Despite the moderate resolution, the quality of the map around the side chain regions is well-defined. We have included the map densities around these residues in the pore region and also for the complete PANX3 molecule in Extended Data Fig. 3. We have displayed the figure panels with greater clarity. Please refer to modified figures 2 and 3 for residue densities.

EDFig. 3b shows density for a lipid. What kind of structure does this density belong to? Why did the authors use POPC to model this density? Did they add POPC during purification and forgot to mention this?

Response: The density resembles a phospholipid (POPE) that could have been extracted along with the channel during extraction. The density matches well with the map and coincides with the phospholipid. Structures previously done for PANX1 show the presence of multiple lipids and detergent densities around the heptamer and also near the protomer interface. We did not add any additional lipids and used only GDN for extraction.

Page 5, first paragraph. "The negatively charged surface may facilitate the role as a calcium channel". What about the positively charged surface in the same pore? May it prevent PANX3 to play the role of calcium channel? The authors need to run MD simulations to decide on which of the surfaces is more important for permeation.

Response: We have removed the statement as we do not have experimental evidence to conclude the role of positively charged surface in calcium efflux and binding in PANX3.

Page 5, second paragraph. The authors claim that the "C-terminus is unlikely to serve as a plug to block access to the channel" and "unlikely to play a role in channel opening". The authors have to create the C-terminally truncated mutant (d373-392) and prove their statements functionally.

Response: We have modified the statement but if needed we could try this experiment in the revision of the manuscript.

Page 6, 3rd paragraph. "The TM1-TM2 linker adopts alpha-helical conformation in PANX3, instead of a loop in PANX1". This is not at all clear from Fig. 2d. Besides, this statement contradicts EDFig. 4a!

Response: We have corrected the figure for better clarity.

“.....”

Fig. 2g. What is the shown current normalized to?

We are now showing the current density plot instead of normalized current-voltage plot for better clarity.

Nonetheless, previously, we had used the highest value as 1 and performed normalization of other values accordingly.

Page 7, second paragraph. What does it mean: voltage-sensitive currents at positive potentials? Currents are always sensitive to voltage according to the Ohm's law!

Response: We regret the confusion. The sentence is modified.

Page 8, second paragraph. "W74 shifts by ... an angle". What does this mean?

Response: The W74 rotamer rotates around the χ^2 -torsion angle of 87 degrees to position the aromatic ring closer to the pore. **(Figure 3a)** .

Figure 3f. The concentration dependence does not show saturation at high concentrations. The concentration range should be extended to high concentrations for fitting to be done correctly.

Response: We optimized the ATP concentration for the MST experiments with respect to PANX1_{WT}. The highest ATP concentration of 2mM showed a proper saturation and a consistent binding affinity within three independent replicates for PANX1_{WT}. Similar conditions were used for the mutants to rule out different variables that might play a role at higher concentrations of ATP- γ S.

For example, at higher concentrations of ATP, we observed a significant change in the pH, most likely due to hydrolysis.

However, if required, we can attempt to repeat the experiments with higher concentrations of ATP to get a proper saturation during the revision of the manuscript.

As pointed out by the reviewer, a proper saturation did not reach to determine the binding affinity for the PANX1_{R217H} mutant accurately, therefore, we had implied it as an apparent affinity in the manuscript that shows altered behaviour of the mutant compared to PANX1_{WT}.

As far as the manuscript and data presentation is concerned, at times, the chosen illustrations do not help the reader at all to interpret the data, nor do they support the written text in the manuscript. I usually do not comment on the style of illustrations, however, in this case some of the figures simply do not support the written text, an

assessment of which becomes therefore unfeasible without access to density maps and corresponding models. I am providing some details below:

This sentence "Two disulfide bonds stabilize the extracellular domains, C66-C261(SS1) and C84-C242(SS2), that form between EL1 and EL2 (Extended Data Fig. 4a, b)" uses too many undefined abbreviations and it is not clear what the authors are trying to say or show here. I would suggest to add a panel to Extended Data Fig. 4 to visualize these interactions and also to rephrase this sentence with clear explanations (what are EL1 and EL2, for instance?).

Response: The modifications have been incorporated in the Extended figure 4b.

The modified text now read as follows:

"The topology of PANX3 protomers is similar to PANX1_{WT}, with differences in the TM1 and the extracellular loops. The topology comprises four transmembrane helices and two extracellular domains. Two disulfide bonds(SS1 and SS2) stabilize the extracellular domains, C66-C261(SS1) and C84-C242(SS2), that form between extracellular loops (EL), EL1 and EL2 (Extended Data Fig. 4a, b)"

"N-linked glycosylation at the N255 position in the EL2 of PANX1 was implicated in preventing the formation of gap junctions." Reference(s) needed here.

Response: Two references have been added that show evidence of the role of N-glycosylation in preventing gap-junction formation on page 5, para 2.

No 16.

Ruan, Z., Orozco, I.J., Du, J. & Lu, W. Structures of human PANX 1 reveal ion pathways and mechanism of gating. *Nature* 584, 646-651 (2020).

No 27.

Boassa, D. et al. PANX1 channels contain a glycosylation site that targets the hexamer to the plasma membrane. *J Biol Chem* 282, 31733-43 (2007).

The authors claim that they observe density for N-acetylglucosamine (NAG) at the predicted glycosylation site in their PANX3 structure. The corresponding panel in Fig 1f. is not self-evidently showing this. Without access to the maps, it is hard to make a judgement here. I would suggest to show an additional close-up of density with the modelled molecule inside it (like POPE in Extended Data Fig. 3 for instance).

Response: The density for N-glycosylation is included in the new extended data figure 5a. Similarly POPE density is also provided as an inset in Extended data fig. 5c.

"A comparison with the Alphafold2 model of PANX3 with the experimental model in this study indicates that the N-terminus faces the cytosol." I think the authors

assume a good PANX structural knowledge from the future readers here. The mentioned Alphafold2 models need to be shown in the manuscript in comparison with the presented structural models. Please either add a figure in Extended Data or clearly refer to an existing figure such as Fig 1d.

Response: As per the reviewers' recommendation, we have added a comparison of the PANX3 experimental structure vs. PANX3 AlphaFold2 model and also a comparison of PANX1 with the PANX2 AlphaFold2 model, which was truncated after residue 376 as the extended C-terminus in PANX2 is predicted to be highly disordered. (Extended Data Fig. 5d-e)

Regarding the first two paragraphs/sections of "PANX3 displays a double-sieve pore organization": the residue depiction in Fig. 2a is unfortunate, especially if cation- π interactions are discussed here, please provide a better, more detailed illustration (no need to show the whole channel here, Fig. 1 showed this already but side chains and densities for them should). Same applies to the salt bridge and hydrogen bond discussed next; please provide an illustration. The cation-pi interaction is not "altered" by W74I, it is "eliminated" because I74 does not have pi-orbitals!

Response: A more detailed illustration of the electrostatics around the upper vestibular chamber is provided along with the residues at the pore (I74) and the boundary (F58) between the upper and lower parts of the vestibule are clearly provided in Fig.1d inset.

The same problem arises with POPE: where is this lipid situated in the map/model? It is completely out of context if no structural/spatial reference is given.

Response: We have altered the figure and provided a clear context as to where lipid interacts at the interface of the PANX3 heptamer in Extended Data figure 5c.

Fig. 2a,b: it took me some time to realise that these panels relate to each other.

The interactions that the authors are trying to discuss are not facilitated at all by the way Fig. 2d is currently designed/depicted. Please consider to show more details/residue interactions here in correspondence with the text.

Response: Our apologies for this confusion. In the newly revised figures (fig. 2), we have altered the panels for greater clarity and provided supporting data in the extended data. I hope the reviewer finds it easier to follow the revised figures.

Again here: "The R75 residue interacts with the D81 of the adjacent subunit through a salt bridge, consistent with PANX1WT." -> where am I supposed to look? No figure provided.

Response: Please refer to Fig. 2d, e for these changes. We have clarified these interactions in newly made panels.

I would mention the 42% sequence identity between PANX1 and 3 in the introduction already, if possible, to provide context for the 27% sequence identity between PANX1 and 2 which is discussed after that.

Response: The sequence identity is now mentioned in the introduction.

Reviewers' comments:

Reviewer #1 (Remarks to the Author):

The authors have addressed my concerns on the initial submission. The manuscript is toned down, eliminating the previously concerning message on dynamics.

It is a tricky topic to work on. However, the authors did the best they could to address mechanisms.

The only specific comment I have is on Fig. 3d.

Is the change in current density a result of different cell surface expressions? An attempt to roughly estimate the relative surface expression level by biotinylation/western blotting or other methods may help.

Reviewer #2 (Remarks to the Author):

The revised manuscript by Hussain et al addresses some of the critical comments. The manuscript still contains valuable information for the field, and I appreciate that the authors made an effort to improve the text and figures. However, the revised manuscript falls somewhat short of what I was hoping to see. I outline below the specific issues throughout the manuscript which I found to be in need of improvement. I hope the authors will find these comments helpful for preparing a new version of the manuscript.

The most immediately obvious shortcoming is that this manuscript still reads as three very loosely connected stories: one about the PANX3, one about PANX1-R217H, and one about the PNAX1-DM mutant (originally meant to mimic the PANX2 channel). There is very little logical connection between these three parts. The rationale for the whole study and for each of its parts is not well explained, and the reader is confronted with the structures, binding assays and electrophysiology data, often stated to show a certain result but with no clear interpretation. One is left to imagine what all this might mean, other than a bunch of facts about different PANX homologues and their mutants. This makes this current version of the manuscript a difficult read. We are missing a red thread that links the three parts of the manuscript - I commented about this previously, and the authors unfortunately have not yet found a way to fix this glaring issue. Perhaps the authors can derive some inspiration from their own text in on page 11, where they speculate on the H-bond network within TM regions, as found in the R217H mutant - I

had a vague feeling that somewhere there might lie the glue that could hold the whole story together. But the current version is not there yet.

Abstract. A relatively clunky first sentence - are the authors trying to state that these channels form hemichannels only, but not the gap junction channels? The second sentence gives some details, but the rationale for the study is not explained. The last sentence of the abstract is relatively unspecific - it is not clear whether these results have any meaning for understanding the mechanisms of action of these channels or their role in pathology. Overall, the abstract in the current form fails to explain the importance of the topic, the relevance of the congenital mutant (and other mutants) and the overall significance of all results.

Throughout - "Pannexin1" or "Pannexin 1"? Spaces are occasionally missing throughout the text.

Throughout - "germline mutant" or "congenital mutant"?

Line 77 - "upon creating PANX1 knock outs" - strange sentence, should be rephrased.

Line 85 - "In this study we present" - as in the abstract, the rationale for the study is not explained. The authors go from introducing the basic facts about the channels straight to the results. Why they decided to study these channels - and these specific constructs - is not explained. This contributes to the feeling that there are several disconnected stories about PANX3 and PANX1 wt/mutants.

Line 99 - "Pannexin3 has altered structural features" - compared to what?

Line 101 - "disparity" - not sure this is the right word here. Did the authors want to understand better the disparity between the sequences of PANX3 and PANX1, and so they decided to solve these structures?

Line 107 - "to model a bulk" - probably can be rephrased

Line 112 - where did PANX1 come from? The PANX1 cryo-EM analysis is not mentioned before this statement.

Line 134 - "the N-terminus faces the cytosol" - in which model? Do you mean to say that the AlphaFold model shows the N-terminus predicted to be extended to the cytosol? Not sure one can claim that this is what actually happens, especially if the cryo-EM map shows no ordered features there.

Line 141 - "However, ATP." - what does this mean and how does this related to the previous sentence? The sentence after does not seem to be logical, at least the first part of it where the authors state that the C-terminus is unlikely to play a role in channel opening. Not clear yet how this conclusion is derived.

Line 156 - "Apart from these interactions.." - this is an opportunity to highlight the interface in a figure panel.

Line 178 - Fig. 1d - is this the right figure?

Line 182 - "We performed a binding analysis.." - comes out somewhat unexpectedly in the middle of this paragraph.

Line 191 - "Moreover, inconsistencies.." - maybe it would be better to not try to talk about the lack of understanding based on failed experiments, which I assume are not going to be shown.

Line 197 - "..cells20,27" - are these references to the failed attempts? Or do you mean that only in the current you could do this analysis. Currently not clear.

Line 215 - No connection to the previous text. Gives an impression of a new direction of the manuscript, not logically linked to the previous part.

Line 218 - "The arginine was mutated.." - move to Methods.

Line 225 - Stating the obvious. Maybe it is possible to rephrase.

Line 231 - "The R75 residue.." - where is the illustration?

Extended Data Fig 6 - labels in C and D are missing. Colours of the curves in the figure do not seem to be consistent.

Line 262 - referring to K24 and R128 - where are these residues? Is there an illustration? These abstract descriptions make it difficult to follow the text.

Line 282 - Ext. Data Fig. 8f, Fig. 3g - why are the same panels duplicated? This adds extra bulk to the manuscript, but does not help in any way.

Line 288 - are the authors claiming a smaller pore size in R75A? Is this a case for solving a cryo-EM structure for R75A mutant?

Line 296 - the double mutant description / rationale is not very clear in the new version of the manuscript. The authors stay away now from PANX2, but this makes the argument for analysing this mutant less clear. Maybe this can be improved still.

Line 300 - PANX1DM - a matter of personal preference, but it may be useful to rebrand the mutant as PANX1dm, or PANX1-WR (otherwise the letters D and M may be suggestive of Asp-Met). But this is a minor point.

Line 304 - "PANX1DM channel whose structure of PANX1DM" - rephrase

Line 324 - interaction with CBX?

Discussion - I think the authors can improve the discussion, avoiding a repetition of their results and trying to connect the three parts together. This is essential for this manuscript to give an impression of a congruent story. Maybe there is a way to derive conclusions about the general properties of the PANX channels, instead of focusing on any one isoform.

Figure 2b and c - the rationale of duplicating the views is not clear, as the same message can be conveyed with either of the panels that are shown.

Figure 2d and e - the panels can be merged.

Figure 2i - what is the scale for the non-transfected cells? It would help to have an equal scale for all three examples (1nA of equal length)

Figure 2j-k - the data for non-transfected cells (Ext. Data Fig. 7?) should be added.

Fig 2j-k and Ext. Data Fig. 7 - using duplicated panels makes no sense. Again this just inflates the manuscript without providing any value.

Fig. 3b - this could go to supplementary (cryo-EM density features), as it shows redundant information as in 3a.

Finally, the amount of data shown per structure corresponds to ~2h of microscope time today. Despite all known limitations, the authors in principle should be able to improve the resolution of their structures substantially by adding a modest amount of additional cryo-EM data.

Reviewer #3 (Remarks to the Author):

My major concern remains the quality of the presented functional data, in particular in view of the high errors and bad statistics. The authors got rid of the error bars and show single readings now. This is not how you improve stats and reduce measurement errors (by removing them).

The authors have not improved the resolution and quality of their maps since the last revision. In particular, I proposed to the authors to attempt focussed classification on the ICD (and preferably also the TMD). This would be one way to address the drastic worsening of density quality in this region. Another reviewer suggested to collect more data which the authors have refused to do. Several speculative statements made by the authors, as highlighted in my previous report, rely on the quality of the maps and the confidence of their interpretation. If invited for revision, I would strongly suggest the authors to improve the quality of their cryo-EM reconstruction by suggested approaches (collecting more data and using alternative processing strategies) and further tone down/remove entirely some of their interpretations.

The authors claim that the "C-terminus is unlikely to serve as a plug to block access to the channel" and "unlikely to play a role in channel opening". The authors have to create the C-terminally truncated mutant (d373-392) and prove their statements functionally.

On a positive site, the authors did improve the representation of their findings, making the manuscript text easier to follow. Nevertheless, several places in the manuscript remain difficult to read and understand so I would encourage the authors to further continue working on their text quality.

Overall, I do believe that this manuscript can be invited for a revision but the authors should put more significant effort in addressing the reviewers' critics than described in their plan of revision.

Responses to Reviewers' comments:

Reviewer #1 (Remarks to the Author):

The authors have addressed my concerns on the initial submission. The manuscript is toned down, eliminating the previously concerning message on dynamics.

It is a tricky topic to work on. However, the authors did the best they could to address mechanisms.

The only specific comment I have is on Fig. 3d.

Is the change in current density a result of different cell surface expressions? An attempt to roughly estimate the relative surface expression level by biotinylation/western blotting or other methods may help.

Response: We thank the reviewer for extending support and we have characterized the surface expression of all the constructs used in the study using HEK293 cells used for electrophysiology through confocal experiments. As represented below the expression levels are similar among most of the constructs suggesting that the current densities measured are unlikely to be influenced by surface expression. Differences in the levels of specific constructs are mentioned in the manuscript and quantified in the image below.

Reviewer #2 (Remarks to the Author):

The revised manuscript by Hussain et al addresses some of the critical comments. The manuscript still contains valuable information for the field, and I appreciate that the authors made an effort to improve the text and figures. However, the revised manuscript falls somewhat short of what I was hoping to see. I outline below the specific issues throughout the manuscript which I found to be in need of improvement. I hope the authors will find these comments helpful for preparing a new version of the manuscript.

The most immediately obvious shortcoming is that this manuscript still reads as three very loosely connected stories: one about the PANX3, one about PANX1-R217H, and one about the PANX1-DM mutant (originally meant to mimic the PANX2 channel). There is very little logical connection between these three parts. The rationale for the whole study and for each of its parts is not well explained, and the reader is confronted with the structures, binding assays and electrophysiology data, often stated to show a certain result but with no clear interpretation. One is left to imagine what all this might mean, other than a bunch of facts about different PANX homologues and their mutants. This makes this current version of the manuscript a difficult read. We are missing a red thread that links the three parts of the manuscript - I commented about this previously, and the authors unfortunately have not yet found a way to fix this glaring issue. Perhaps the authors can derive some inspiration from their own text in on page 11, where they speculate on the H-bond network within TM

regions, as found in the R217H mutant - I had a vague feeling that somewhere there might lie the glue that could hold the whole story together. But the current version is not there yet.

Response: We thank the reviewer for a thorough and constructive review and do agree that the writing style in the earlier version suggested disconnected stories. In the revised version we have put in substantial effort to improve the flow of the manuscript and have a logical connection between the different results of the manuscript. Consequently, aided by the recent publication of the PANX2 structures, this study constitutes a rather extensive analyses into the comparison of the PANX isoforms both from structural and functional perspective. It further delves in to the roles of the pore and vestibule lining residues through mutagenesis and functional analyses as we added more data on the substitutions of the PANX3 pore and C-terminus deletion to facilitate a better understanding of PANX3 and its comparison to the other isoforms. We earnestly hope that this version of the manuscript is satisfactory to the reviewer.

Abstract. A relatively clunky first sentence - are the authors trying to state that these channels form hemichannels only, but not the gap junction channels? The second sentence gives some details, but the rationale for the study is not explained. The last sentence of the abstract is relatively unspecific - it is not clear whether these results have any meaning for understanding the mechanisms of action of these channels or their role in pathology. Overall, the abstract in the current form fails to explain the importance of the topic, the relevance of the congenital mutant (and other mutants) and the overall significance of all results.

Response: The abstract is now altered as per the suggestions of the reviewer wherein we improved the clarity and states the following, “Pannexins are single-membrane large-pore ion channels that release ATP upon activation. Three isoforms of pannexins, 1, 2, and 3, perform diverse cellular roles and differ in their pore lining residues. In this study, we report the cryo-EM structure of pannexin 3 at 3.9 Å and analyze its structural differences with pannexin isoforms 1 and 2. The pannexin 3 vestibule has two distinct chambers and a wider pore radius in comparison to pannexins 1 and 2. We further report the cryo-EM structures of pannexin 1 pore substitutions W74R/R75D that mimic the pore lining residues of pannexin 2 and a germline mutant of pannexin 1, R217H at resolutions of 3.2 Å and 3.9 Å, respectively. The germline mutant R217H in transmembrane helix 3 (TM3), leads to a partially constricted pore, reduced ATP interaction propensities and weakened voltage sensitivity. Substitution of cationic residues in the vestibule of pannexin 1 results in reduced ATP interaction propensities to pannexin 1. The study compares the three pannexin isoform structures, the effects of substitutions of pore and vestibule-lining residues and allosteric effects of a pathological substitution on channel structure and function thereby enhancing our understanding of this group of ATP-release channels.”

Throughout - “Pannexin1” or “Pannexin 1”? Spaces are occasionally missing throughout the text.

Response: We fixed this spacing and made sure the spacing is there for all the words.

Throughout - “germline mutant” or “congenital mutant”?

Response: We have altered the phrase to “germline mutant” in the entire manuscript as per the original manuscript where this mutant is reported.

Line 77 - “upon creating PANX1 knock outs” - strange sentence, should be rephrased.

Response: The sentence is rephrased as follows, “PANX3 is suggested to functionally compensate for the absence of PANX1 in the vomeronasal organ of PANX1 knock out mice”

Line 85 - “In this study we present” - as in the abstract, the rationale for the study is not explained. The authors go from introducing the basic facts about the channels straight to the results. Why they decided to study these channels - and these specific constructs - is not explained. This contributes to the feeling that there are several disconnected stories about PANX3 and PANX1 wt/mutants.

Response: We thank the reviewer for this suggestion. The following sentences have been introduced in front of Line 85 that clarify the context of performing this study. “Incidentally the PANX3 has a branched hydrophobic amino acid, isoleucine (I74), at this position which is a tryptophan (W74) in PANX1 and cationic arginine (R89) in PANX2. These isoform specific changes at the pore can influence channel properties and the study aims to explore the structural and functional effects of these pore lining residues in pannexin isoforms 1 and 3.”

Line 99 - “Pannexin3 has altered structural features” - compared to what?
Response: The title of this section is modified to the following, “Pannexin 3 has distinct structural features in comparison to other isoforms”.

Line 101 - “disparity” - not sure this is the right word here. Did the authors want to understand better the disparity between the sequences of PANX3 and PANX1, and so they decided to solve these structures?

Response: We modified the phrase as follows, “To understand the architectural differences among PANX isoforms.”

Line 107 - “to model a bulk” - probably can be rephrased

Response: Changed to “model most of the side chains”.

Line 112 - where did PANX1 come from? The PANX1 cryo-EM analysis is not mentioned before this statement.

Response: We had performed a reconstruction of PANX1_{WT} at a moderate resolution but there were several studies with published PANX1 structures in the intervening time. We therefore resorted to the use of PANX1 coordinates from a study where the channel has been reported at the highest resolution and purified in the same detergent and buffer conditions. The coordinates also match well to our map and therefore we submitted only the map and not the coordinates for this reconstruction.

Line 134 - “the N-terminus faces the cytosol” - in which model? Do you mean to say that the AlphaFold model shows the N-terminus predicted to be extended to the cytosol? Not sure one can claim that this is what actually happens, especially if the cryo-EM map shows no ordered features there.

Response: We have addressed the reviewer's comment regarding the AlphaFold2 model of PANX3, where N-terminus is predicted to be towards cytosol. As highlighted by the reviewer, our determined structure does not exhibit any density towards the N-terminus. Consequently, we have omitted these statements to enhance the clarity of the manuscript.

Line 141 - “However, ATP..” - what does this mean and how does this relate to the previous sentence? The sentence after does not seem to be logical, at least the first part of it where the authors state that the C-terminus is unlikely to play a role in channel opening. Not clear yet how this conclusion is derived.

Response: We initially speculated that the C-terminus of PANX3 does not influence channel gating due to its small size. However, in response to the concerns raised by the reviewers, we generated a C-terminal deletion mutant in PANX3 (PANX3_{C-del}) and conducted whole-cell patch clamp recordings. Intriguingly, our observations revealed no discernible difference in current density between the presence and absence of the C-terminus in PANX3. Additionally, our assessment of surface expression indicated that PANX3 C-terminus deleted construct exhibits higher expression levels compared to PANX3_{WT}, suggesting that the C-terminus does not function as a pore blocker, in contrast to PANX1 where the C-terminus has been demonstrated to act as a pore blocker.

Line 156 - “Apart from these interactions..” - this is an opportunity to highlight the interface in a figure panel.

Response: The following figure panel was introduced in Extended Data Figure 6e.

Line 178 - Fig. 1d - is this the right figure?

Response: The numbering for the figure is correct. However, we have now added PANX2 structure for comparison among PANX isoforms.

Line 182 - “We performed a binding analysis..” - comes out somewhat unexpectedly in the middle of this paragraph.

Response: We start a new paragraph here as follows,

“Given the role of PANX 1 and 3 isoforms in ATP release activity^{9,30}, we analysed ATP interactions with the channel through binding analysis of ATP- γ S with PANX isoforms utilizing microscale thermophoresis (MST).”

Line 191 - “Moreover, inconsistencies..” - maybe it would be better to not try to talk about the lack of understanding based on failed experiments, which I assume are not going to be shown.

Response: We have removed the sentence from the text.

Line 197 - “..cells20,27” - are these references to the failed attempts? Or do you mean that only in the current you could do this analysis. Currently not clear.

Response: References 24, 25 have been moved to the earlier part of the sentence signifying previous failed attempts.

Line 215 - No connection to the previous text. Gives an impression of a new direction of the manuscript, not logically linked to the previous part.

Response: The following statement was added to link the section 2 with section 3. The manuscript has been reordered at this stage and section “PANX1 pore substitutions mimic pore residues of PANX2” is now included at this stage with the section on congenital mutant moved to the end of results section. Some lines were added in the previous section as follows to connect to the subsequent section, “A notable difference between PANX1 and PANX3 is the pore-lining residue, I74 in PANX3 as opposed to W74 in PANX1. To unravel the consequences of this difference, we substituted I74 to a tryptophan (I74W) and alanine (I74A), respectively and observed the impact of these alterations on channel function in PANX3. We observe that substitutions at this site are not tolerated and lead to a loss of any measurable current and a reduced surface expression among PANX3 substitutions (Figure 2m, Extended Data Figure 6b). This is unlike PANX1 where W74I substitution retained PANX1_{WT} like currents²⁵.”

Line 218 - “The arginine was mutated..” - move to Methods.

Response: We have shifted the sentence to the methods section. The section is also relocated in the manuscript.

Line 225 - Stating the obvious. Maybe it is possible to rephrase.

Response: We have rephrased this paragraph which starts as follows, “Despite the lower resolution of the structure, it resembles PANX1_{WT} globally, with subtle changes in extracellular domain (ECD), extracellular helix 1 (EH1), and the intracellular domain (ICD).”

Line 231 - “The R75 residue..” - where is the illustration?

Response: We have now put an illustration in the figure 5 (Main Figure 5b).

Extended Data Fig 6 - labels in C and D are missing. Colours of the curves in the figure do not seem to be consistent.

Response: We have addressed and rectified the issues by implementing the necessary changes to the figure panels. The mentioned data is now part of Extended Data figure 8 and main figure 4f.

Line 262 - referring to K24 and R128 - where are these residues? Is there an illustration? These abstract descriptions make it difficult to follow the text.

Response: We have incorporated a figure depicting the position of all the mutants studied in the manuscript (Main Figure 3a). We have also performed electrophysiology recording for these two mutants.

Line 282 - Ext. Data Fig. 8f, Fig. 3g - why are the same panels duplicated? This adds extra bulk to the manuscript, but does not help in any way.

Response: Duplicated panels have been removed.

Line 288 - are the authors claiming a smaller pore size in R75A? Is this a case for solving a cryo-EM structure for R75A mutant?

Response: In the latest version of the manuscript, we clearly state that PANX1_{R75A} mutant, given its interactions with W74 and with the D81 of adjacent protomer, could create a destabilized pore. The text is written as follows in the revised manuscript and we do not plan to solve a structure of this mutant, “Moreover, electrophysiology data shows a two-fold decrease in the current density in PANX1_{R75A} compared to PANX1_{WT}, although surface expression for the mutant is significantly higher than the PANX1_{WT}. Since the residue is involved in stabilizing interactions with W74 and interprotomeric interactions, R75A substitution could locally alter the interactions around the pore. Further, normalized G-V curve for R75A is comparable to wild type and reveals that the substitution of a pore residue, R75, although alters the channel properties but does not affect the voltage sensitivity of the channel (Figure 4f-g) consistent with the findings with ZfPANX1³⁶.”

Line 296 - the double mutant description / rationale is not very clear in the new version of the manuscript. The authors stay away now from PANX2, but this makes the argument for analysing this mutant less clear. Maybe this can be improved still.

Response: This section is now reordered and we state the rationale for this mutant as follows, “As observed in the PANX3_{I74W} mutant, a single mutation at the pore led to an inactive channel. Moreover, single substitutions of W74R and R75D in the pore region yielded incorrectly assembled PANX1²⁶. In order to observe if a dual substitution of both the residues (W74, R75) by charged amino acids could compensate for this behavior, we created a PANX1 double mutant (PANX1_{WR/RD}) by substituting W74 and R75 with arginine and aspartate, respectively (Figure 3a).”

Line 300 - PANX1DM - a matter of personal preference, but it may be useful to rebrand the mutant as PANX1dm, or PANX1-WR (otherwise the letters D and M may be suggestive of Asp-Met). But this is a minor point.

Response: We now call PANX1_{DM} as PANX1_{WR/RD} which makes it easier to understand.

Line 304 - “PANX1DM channel whose structure of PANX1DM” - rephrase
Response: The sentence has been rephrased to the following, “we obtained a minor fraction of well assembled PANX1_{WR/RD} particles that allowed reconstruction of the PANX1_{WR/RD} structure to a resolution of 3.2 Å.”

Line 324 - interaction with CBX?

Responses: We modified the sentence

Discussion - I think the authors can improve the discussion, avoiding a repetition of their results and trying to connect the three parts together. This is essential for this manuscript to give an impression of a congruent story. Maybe there is a way to derive conclusions about the general properties of the PANX channels, instead of focusing on any one isoform.

Response: We have improved the discussion by focusing on the three major results from this study and removing statements that may indicate a reiteration of results.

Figure 2b and c - the rationale of duplicating the views is not clear, as the same message can be conveyed with either of the panels that are shown.

Response: Duplicating views are now removed in figure 2.

Figure 2d and e - the panels can be merged.

Response: The figure panels are now merged.

Figure 2i - what is the scale for the non-transfected cells? It would help to have an equal scale for all three examples (1nA of equal length)

Response: The scale of 1 nA has been added for untransfected cells. Similarly, a new panel for control experiments with untransfected cells is added. (Main Figure 2j)

Figure 2j-k – the data for non-transfected cells (Ext. Data Fig. 7?) should be added.

Response: Added, as Figure 2j.

Fig 2j-k and Ext. Data Fig. 7 – using duplicated panels makes no sense. Again this just inflates the manuscript without providing any value.

Response: The duplicated panels have been removed.

Fig. 3b – this could go to supplementary (cryo-EM density features), as it shows redundant information as in 3a.

Response: The panel 3b is now merged with panel 3a.

Finally, the amount of data shown per structure corresponds to ~2h of microscope time today. Despite all known limitations, the authors in principle should be able to improve the resolution of their structures substantially by adding a modest amount of additional cryo-EM data.

Response: While we agree that the cryoEM data requested amounts to a short duration on the microscope, we have had severely hampered access to a decent microscope for large data collection over the last one year and the improvements done here were facilitated by an overseas facility that gives very limited time. Despite the challenges we have improved the resolution of the PANX1_{WR/RD} mutant to 3.2 Å. We also tried to improve the PANX 3 structure and did collect data at ESRF cm01. However, we could obtain a resolution of 4.4 Å and therefore chose to not report that data for PANX3 and go with the previous map and structure at 3.9 Å. For PANX1 R217H mutant we performed local refinement as suggested by reviewer 3 and obtained a marginally improved map of 3.77 Å for the transmembrane region although we were unsuccessful in

improving the resolution of ICD domain (Intracellular domain) most likely due to its small size. We therefore sought to work with the earlier map. The improved structure for PANX1_{WR/RD} is updated in the pdb and supplied along with the manuscript as source data.

Reviewer #3 (Remarks to the Author):

My major concern remains the quality of the presented functional data, in particular in view of the high errors and bad statistics. The authors got rid of the error bars and show single readings now. This is not how you improve stats and reduce measurement errors (by removing them).

Response: We regret reporting a single graph for the MST data. We have therefore followed a standard practise to normalise the three recordings and present the data in an improved form rather than averaging raw data. The errors as you will notice are lower compared to the earlier report.

The authors have not improved the resolution and quality of their maps since the last revision. In particular, I proposed to the authors to attempt focussed classification on the ICD (and preferably also the TMD). This would be one way to address the drastic worsening of density quality in this region. Another reviewer suggested to collect more data which the authors have refused to do. Several speculative statements made by the authors, as highlighted in my previous report, rely on the quality of the maps and the confidence of their interpretation. If invited for revision, I would strongly suggest the authors to improve the quality of their cryo-EM reconstruction by suggested approaches (collecting more data and using alternative processing strategies) and further tone down/remove entirely some of their interpretations.

Response: In this version we have improved the resolution for PANX1_{WR/RD} from 4.3 Å to 3.2 Å. Additionally, data for PANX3 was collected at CM01, ESRF. As observed in supplementary figure 1, PANX3 channel has a non-uniform distribution, resulting in the acquisition of only 17K intact particles, leading to a 4.4 Å structure, which is considerably lower than the existing structure.

We also performed focussed refinement for the transmembrane domain(TMD) and intracellular domain for PANX3 but were unsuccessful in achieving a better resolution. We want to emphasize that we have taken measures to display side chain density when drawing conclusions based on it. Furthermore, we have removed any speculative statements.

After undergoing focussed refinement, we have achieved a modest improvement in the resolution of PANX1_{R217H}, specifically in the Extracellular Domain (ECD) and the transmembrane region. However, we were unsuccessful in improving the resolution of ICD domain (Intracellular domain). To emphasize the reliability of the previously built model specifically at W74 region, we have included a corresponding figure (Extended Data Figure 9c).

The authors claim that the "C-terminus is unlikely to serve as a plug to block access to the channel" and "unlikely to play a role in channel opening". The authors have to create the C-terminally truncated mutant (d373-392) and prove their statements functionally.

Response: We have created C-terminus deletion of PANX3 and observe that there is no significant change in the current densities measured. We therefore report the not so critical role of the C-terminus in PANX3 as compared to PANX1 isoform.

On a positive site, the authors did improve the representation of their findings, making the manuscript text easier to follow. Nevertheless, several places in the manuscript remain difficult to read and understand so I would encourage the authors to further continue working on their text quality.

Response: We do appreciate the feedback by the reviewer on text quality. We have added substantial amount of new and relevant results and altered the writing organization and style for an improved flow of the manuscript. I hope this revised version would be acceptable to the reviewer.

Overall, I do believe that this manuscript can be invited for a revision but the authors should put more significant effort in addressing the reviewers' critics than described in their plan of revision.

Response: We thank you for the opportunity to revise the manuscript. As mentioned earlier we have put a substantial effort in improving the quality of the reported structures and data quality within the manuscript.

REVIEWERS' COMMENTS

Reviewer #1 (Remarks to the Author):

The authors have satisfactorily addressed my primary concern regarding the cell surface expression level of the mutants. However, a few points could be clarified or expanded upon (without requiring additional experiments) to enhance the depth of the manuscript.

1. Line 18-19

Pannexins are single-membrane large-pore ion channels that release ATP upon activation.

This does not make sense.

Pannexins are large-pore channels that release ions and ATP.

2. Line 179-181: The discussion on the lipid's role in pannexins (Panx 1-3) could be expanded. Is this lipid conserved across all Panx variants? Could there be functional regulation by a tightly bound lipid, similar to the findings in CALHM1 as reported by Syrjanen et al. in their 2023 Nature Communications paper? Elaborating on this could significantly enrich the manuscript.

3. Line 183-185: The side-tunnel hypothesis in the context of Panx1 is somewhat controversial, mainly because another phospholipid reportedly obstructs this site. Can the authors confirm the presence of a lipid density around this site in their study? Or does the resolution and size of the data not permit such observation?

4. ATP-gS Binding Assay: The implications of the assay results are somewhat unclear. Do they suggest that ATP-gS binds and blocks the channel at high concentrations? Or, are the authors proposing a mechanism where binding and unbinding of ATP-gS plays a role in the specific permeation of ATP through the channel?

Clarifications and additional discussions on these points would greatly enhance the manuscript's comprehensiveness and depth.

Reviewer #2 (Remarks to the Author):

The revised manuscript by Hussain et al has been improved, and I think the authors made a decent effort to address the various criticisms. I have only a few points:

1. The last sentence of the abstract - too unspecific what the conclusions from all this work are.
2. The argument that the authors could not get more cryo-EM time is understandable, but it is progressively more difficult to accept as time goes by. The EM time used to be quite limiting a few years back, but by now there should be many EM facilities that would be able to offer beam time - the authors would need to be persistent about finding such opportunities and likely this would require an extra effort. Clearly for the purpose of this study the authors decided to abandon the strategy of collecting more EM data (probably they do not expect to derive any substantial new insights from higher resolution EM maps). However, in case higher resolution maps may produce new insights, I think it would be worth explicitly pointing this out in the manuscript and clearly stating which of the current conclusions may potentially suffer from the current limited resolution.

Responses to reviewers

Reviewer #1 (Remarks to the Author):

The authors have satisfactorily addressed my primary concern regarding the cell surface expression level of the mutants. However, a few points could be clarified or expanded upon (without requiring additional experiments) to enhance the depth of the manuscript.

1. Line 18-19, Pannexins are single-membrane large-pore ion channels that release ATP upon activation. This does not make sense.

Pannexins are large-pore channels that release ions and ATP.

Answer: We thank the reviewer for thorough and constructive review of the manuscript. As suggested, we have incorporated the changes to the abstract.

2. Line 179-181: The discussion on the lipid's role in pannexins (Panx 1-3) could be expanded. Is this lipid conserved across all Panx variants? Could there be functional regulation by a tightly bound lipid, similar to the findings in CALHM1 as reported by Syrjanen et al. in their 2023 Nature Communications paper? Elaborating on this could significantly enrich the manuscript.

Answer: We have included the suggested reference and added the following lines to the section to address the concern of the reviewer. "The phospholipid could be enhancing the interactions between the protomers of PANX3 akin to the phospholipid interactions observed in the CALHM1 channels³⁴.

However, besides this lipid density we did not observe any other prominent density for lipid at the inner leaflet of the bilayer likely due to the lower resolution of this structure compared to other channels among large pore ion channels."

3. Line 183-185: The side-tunnel hypothesis in the context of Panx1 is somewhat controversial, mainly because another phospholipid reportedly obstructs this site. Can the authors confirm the presence of a lipid density around this site in their study? Or does the resolution and size of the data not permit such observation?

Answer: We didn't observe any lipid density at this site. Furthermore, we also didn't observe any lipid density in PANX2 structure at this site (He, Z., Zhao, Y., Rau, M.J. et al. *Nat Commun* (2023). In addition, the PANX1 structures were resolved at a much higher resolution and no density of lipids was observed at this site suggesting that absence of observation of lipids in the PANX3 structure might not be because of the limited size and resolution of the data. We introduced the following sentence in the paragraph in page 7 in response to the reviewer's suggestion. "Although we did not observe a lipid density around this region that could occlude this region, a higher resolution structure of PANX3 might reveal greater details about the solvent accessibility around this portal."

4. ATP-gS Binding Assay: The implications of the assay results are somewhat unclear. Do they suggest that ATP-gS binds and blocks the channel at high concentrations? Or, are the authors proposing a mechanism where binding and unbinding of ATP-gS plays a role in the specific permeation of ATP through the channel?

Clarifications and additional discussions on these points would greatly enhance the manuscript's comprehensiveness and depth.

Answer: With the available data, it is more likely that binding and unbinding of ATP-gS play a role in the specific permeation of ATP through the channel. The following sentence in the discussion is modified to reflect this. "... are likely to influence the ability of the channel vestibule to sequentially bind and unbind ATP resulting in its permeation and release."

Reviewer #2 (Remarks to the Author):

The revised manuscript by Hussain et al has been improved, and I think the authors made a decent effort to address the various criticisms. I have only a few points:

1. The last sentence of the abstract - too unspecific what the conclusions from all this work are.

Answer: Since the abstract summarizes the impact of the study and as per the editor's suggestion, we retained the final sentence as is.

2. The argument that the authors could not get more cryo-EM time is understandable, but it is progressively more difficult to accept as time goes by. The EM time used to be quite limiting a few years back, but by now there should be many EM facilities that would be able to offer beam time - the authors would need to be persistent about finding such opportunities and likely this would require an extra effort. Clearly for the purpose of this study the authors decided to abandon the strategy of collecting more EM data (probably they do not expect to derive any substantial new insights from higher resolution EM maps). However, in case higher resolution maps may produce new insights, I think it would be worth explicitly pointing this out in the manuscript and clearly stating which of the current conclusions may potentially suffer from the current limited resolution.

Answer: We agree with the reviewer, and we did make extensive attempts at gaining higher resolution data. A consequence of this was the improvement of our PANX1_{WR/RD} structure resolution from 4.3 to 3.2 angstroms. Our attempts to improve PANX3 resolution were unsuccessful as we got only a 4.2 angstrom data which was worse than the earlier structure reported at 3.9. We therefore would like to assure the reviewer that all possible attempts were made to improve the data quality with the available choices for this study. We have in multiple instances indicated that improved resolution would aid in providing greater details of the channel function. Some instances are given below.

Page 7; para 2: "However, besides this lipid density, we did not observe any other prominent density for lipid at the inner leaflet of the bilayer likely due to the lower resolution of this structure compared to other channels among large-pore ion channels."

Page 12; para1: "Despite the lower resolution of the structure, it resembles PANX1_{WT} globally, with subtle changes in extracellular domain (ECD), extracellular helix 1 (EH1), and the intracellular domain (ICD). Extracellular loops and EH1 have an average shift of 1.4-1.6 Å away from the pore. The W74 residue of PANX1_{R217H} fits better into the density of its sidechain with a rotameric shift that coincidentally constricts the pore radius".

Page 14; para 2: “Although reported at a lower resolution compared to other structures of PANX1, the mutant PANX1 displays subtle, long-range structural shifts that affect channel function by causing a constriction of pore radius”.